Published in Transations on Machine Learning Research (07/2022)

# Identifying Causal Structure in Dynamical Systems

**Dominik Baumann** *dominik.baumann@it.uu.se*
*Department of Information Technology*
*Uppsala University*
*Uppsala, Sweden*

**Friedrich Solowjow** *friedrich.solowjow@dsme.rwth-aachen.de*
*Institute for Data Science in Mechanical Engineering*
*RWTH Aachen University*
*Aachen, Germany*

**Karl H. Johansson** *kallej@kth.se*
*Division of Decision and Control Systems and Digital Futures*
*KTH Royal Institute of Technology*
*Stockholm, Sweden*

**Sebastian Trimpe** *trimpe@dsme.rwth-aachen.de*
*Institute for Data Science in Mechanical Engineering*
*RWTH Aachen University*
*Aachen, Germany*

**Reviewed on OpenReview:** *https://openreview.net/forum?id=X2BodlyLvT*

## Abstract

Mathematical models are fundamental building blocks in the design of dynamical control systems. As control systems are becoming increasingly complex and networked, approaches for obtaining such models based on first principles reach their limits. Data-driven methods provide an alternative. However, without structural knowledge, these methods are prone to finding spurious correlations in the training data, which can hamper generalization capabilities of the obtained models. This can significantly lower control and prediction performance when the system is exposed to unknown situations. A preceding causal identification can prevent this pitfall. In this paper, we propose a method that identifies the causal structure of control systems. We design experiments based on the concept of controllability, which provides a systematic way to compute input trajectories that steer the system to specific regions in its state space. We then analyze the resulting data leveraging powerful techniques from causal inference and extend them to control systems. Further, we derive conditions that guarantee the discovery of the true causal structure of the system. Experiments on a robot arm demonstrate reliable causal identification from real-world data and enhanced generalization capabilities.

## 1 Introduction

When learning models for dynamical control systems, we ideally would like to obtain models that *(i)* generalize well to new domains, *(ii)* are interpretable, and *(iii)* computationally efficient. However, generalization to new domains and interpretability are particular weaknesses of current black-box machine learning methods (Schölkopf, 2022; Rudin, 2019). We address this problem by first identifying a control system's causal structure and subsequently using this structural knowledge for model learning.

As the understanding of causality differs depending on the domain, we first provide some intuition of what kind of structure we seek to identify. For this, following Eichler (2012), we consider two notions. The first is

*temporal precedence*: causes precede their effects. Temporal precedence is the typical causality notion used in systems theory (Hespanha, 2018, p. 31). However, here, we mainly focus on the second notion, *physical influence*: manipulating causes changes the effects. In other words, we seek experimental routines and tests that enable control systems to learn *(i)* what is the influence of their internal states on one another, and *(ii)* which of their inputs influence which internal states.

Incorporating the causal structure into the model learning problem targets the three characteristics of ideal models outlined above. A key reason for the shortcomings of most current machine learning algorithms concerning those characteristics is that they mostly rely on a pure statistical analysis of the data (Pearl, 2018). Thus, when considering stochastic systems and finite sample sizes, these algorithms will, due to spurious correlations, typically find connections between variables, although those variables do not influence one another based on the underlying physical equations. This can lead to catastrophic errors when extrapolating outside the training data, thus, diminishing generalization capabilities. Further, such models are less interpretable since it is unclear whether connections between variables in the model indicate a causal influence or merely spurious correlation found in the training data. A causal analysis mitigates both problems since it identifies which variables actually have a causal influence on each other. Incorporating this knowledge into the model learning problem yields more interpretable models that generalize better to new domains (Pearl & Mackenzie, 2018). Lastly, when only connections between variables that actually have a causal influence on one another are considered, this also reduces the model's parameter space. Therefore, the causal analysis also helps make the model computationally more efficient.

In this paper, we propose an algorithm that automatically identifies the causal structure of a control system. The problem of inferring causal structure from observational data, i.e., given data for which we cannot influence the data generating process, has been addressed using various methods, see Spirtes et al. (2000); Peters et al. (2017) for an overview. For control systems, solely relying on observational data is not necessary. Such systems are equipped with an input, which they can use to actively conduct experiments for causal inference. Causal inference from experiments, or interventions, has also been studied, most prominently in the context of the do-calculus (Pearl, 1995). However, there it is assumed that state variables can be directly influenced by the input, which is often not possible in control systems. In control systems, it is essential to consider a proper notion of *controllability*, i.e., how the system can be steered to particular regions in the state space through appropriate input trajectories. To the best of our knowledge, such notions have not been considered yet in the causality literature. In the literature on model learning and system identification, we find many techniques for learning mathematical models for control systems (Ljung, 1999; Nguyen-Tuong & Peters, 2011; Schoukens & Ljung, 2019). In system identification, the problem of the exploding number of parameters for black-box methods has, for instance, been addressed using regularization techniques (Schoukens & Ljung, 2019). While these methods can reduce the number of parameters, they may exclude parameters representing a causal influence or include parameters representing spurious correlation depending on the regularization parameter. Thus, the obtained models may fall short on generalization and interpretation capabilities.

**Contributions.** In summary, existing methods from causal inference do not consider a proper notion of controllability, while existing methods from system identification cannot give formal guarantees on finding the true causal structure. In this paper, we bridge this gap and present an algorithm that identifies a control system's causal structure through an experimental design based on a suitable controllability notion and a subsequent causal analysis, for which we can provide formal guarantees. For the causal analysis, we leverage powerful kernel-based statistical tests based on the maximum mean discrepancy (MMD) (Gretton et al., 2012). Since the MMD has been developed for independent and identically distributed (i.i.d.) data, we extend it by deriving conditions under which the MMD still yields valid results and by coming up with a test statistic for hypothesis testing, despite non-i.i.d. data. In terms of controllability, we investigate three different settings: *(i)* exact controllability, where we can exactly steer the system to a desired position, *(ii)* stochastic controllability, where we can only steer the system to an $\epsilon$-region around the desired position, and *(iii)* the special case of linear systems with Gaussian noise that are controllable in the sense of Kalman (Kalman, 1960a), as they represent a widely studied class of systems. We demonstrate the proposed method's applicability by automatically identifying the causal structure of a robotic system and a simulated, nonlinear quadruple tank system. Further, we show improved generalization capabilities for the robotic system

and reduced computational complexity for the quadruple tank system, both inherited through the causal identification.

## 2 Related Work

To the best of our knowledge, no other algorithm seeks to identify the causal structure of a dynamical control system through experiments based on a suitable controllability notion. However, several works in causal inference aim to infer the causal structure of dynamical systems, and several methods in system identification seek to reduce the parameter space or identify structural properties of control systems. In this section, we discuss those works.

**Causal inference for dynamical systems.** Causal inference in dynamical systems or time series has been studied in Demiralp & Hoover (2003); Eichler (2010); Moneta et al. (2011); Malinsky & Spirtes (2018) using vector autoregression, in Peters et al. (2013) based on structural equation models, in Entner & Hoyer (2010), using the fast causal inference algorithm (Spirtes et al., 2000), and in Salvi et al. (2021) and Quinn et al. (2011), applying kernel mean embeddings and directed information, respectively. In Pfister et al. (2019), the authors develop a procedure for learning the causal structure of kinetic systems. A more extensive overview of causal inference methods that can be applied to time-series data, with a particular focus on Earth system sciences, is provided in Runge et al. (2019a). While all these methods make causal inference for dynamical systems, none of them investigates experimental design. Instead, they aim to infer the causal structure from observational data and, thus, need additional assumptions to arrive at statements about the causal structure. Dynamical systems, as considered in this work, can be actively influenced through a control input. Hence, we can design experiments and do not need to rely on the data being sufficiently rich.

**Experimental design.** A well-known concept for causal inference from experiments is the do-calculus. In the basic setting, a variable is clamped to a fixed value, and the distribution of the other variables conditioned on this intervention is studied (Pearl, 1995). Extensions to more general classes of interventions exist, see, e.g., Yang et al. (2018); Shanmugam et al. (2015), but they consider static models, which is different from the dynamical systems studied herein. Causal inference in dynamical systems or time series with interventions has been investigated in Eichler (2012); Peters et al. (2022); Mooij et al. (2013); Rubenstein et al. (2018); Sokol & Hansen (2014). However, therein it is assumed that one can directly manipulate the variables, e.g., by setting them to fixed values or forcing them to follow a trajectory, which is typically impossible in practice. None of those works considers various degrees of controllability, which is the case in this paper. Thus, they are not readily applicable to control systems.

**Model selection and regularization.** As an alternative to directly testing causal relations between variables, several methods exist that identify a dynamic model, trading off model complexity and accuracy. Typically, this is done by letting the algorithm select from a set of candidate models. In system identification, the Akaike information criterion (Akaike, 1973) and the Bayesian information criterion (Schwarz, 1978) are two well-known examples of such methods. In neuroimaging, there are dynamic causal models (Friston et al., 2003; Stephan et al., 2010). A third family of methods are symbolic regression techniques (Bongard & Lipson, 2007; Schmidt & Lipson, 2009; Brunton et al., 2016). In all cases, the true causal structure of the system can only be revealed if a model representing this structure is part of the candidate models. Further, they typically use a regularization parameter to find a trade-off between model complexity and accuracy. This parameter punishes model complexity (e.g., the number of parameters) and rewards goodness of fit. Thus, it also depends on the specific choice of this regularization parameter whether or not these algorithms find a model representing the system's true causal structure.

**Structure detection in dynamical systems.** Revealing causal relations in a dynamical system can be interpreted as identifying its structure. Related ideas exist in the identification of hybrid and piecewise affine systems (Roll et al., 2004; Lauer & Bloch, 2018). These approaches try to find a trade-off between model complexity and fit but cannot guarantee to find the true causal structure. Further methods that identify structural properties of dynamical systems can be found in topology identification (Materassi & Innocenti, 2010; Shahrampour & Preciado, 2014; van den Hof et al., 2013) and complex dynamic networks (Boccaletti et al., 2006; Liu et al., 2009; Yu, 2010). Those works seek to find interconnections between subsystems

instead of identifying a system's inner structure as done herein. Moreover, while the mentioned works often rely on restrictive assumptions such as known interconnections or linear dynamics, our approach can deal with nonlinear systems and does not require prior knowledge.

**Kernel mean embeddings.** For causal inference, we will leverage concepts based on kernel mean embeddings, which have been widely used for causal inference (Peters et al., 2017; Chen et al., 2014; Lopez-Paz et al., 2015). A downside of those methods is that they typically assume that data has been drawn i.i.d. from the underlying probability distributions. Extensions to non-i.i.d. data exist (Chwialkowski & Gretton, 2014; Chwialkowski et al., 2014), but rely on mixing time arguments. Dynamical systems, as investigated in this work, often have large mixing times or do not mix at all (Simchowitz et al., 2018). Therefore, these types of analyses are not sufficient in this case.

## 3 Problem Setting and Main Idea

We consider dynamical control systems of the form

$$x(t) = f(x(0), u(0\!:\!t), v(0\!:\!t)) \tag{1}$$

with discrete time index $t \in \mathbb{N}$, dynamics function $f$, state $x(t) \in \mathcal{X} \subset \mathbb{R}^n$, state space $\mathcal{X}$, input $u(t) \in \mathcal{U} \subset \mathbb{R}^m$, input space $\mathcal{U}$, and $v(t) \in \mathbb{R}^n$ an independent random variable sequence. The notation $(0\!:\!t)$ here denotes the sequence $u(0), u(1), \ldots, u(t)$. In the following, we will omit this notation and simply write $u$ or $v$ if we consider the whole trajectory. The description of the system in equation 1 is different from the standard, incremental version $x(t+1) = \tilde{f}(x(t), u(t), v(t))$. Specifically, equation 1 emphasizes the dependence of $x$ at time $t$ on the initial state. Equation 1 can readily be obtained by iterating $\tilde{f}$ starting from $x(0)$.

We will lend notation from the do-calculus (Pearl, 1995) for causal inference. In this notation, $\mathbb{P}(x(t) \mid \mathrm{do}(x_i(0) = x_i^{\mathrm{I}}(0)))$ defines the conditional probability distribution of $x(t)$ given that we *set* the initial condition of the $i$th component of the state vector $x$ to a specific value $x_i^{\mathrm{I}}(0)$ while all other components of $x$ and the inputs $u$ are left unaffected. In this article, we show how such do-conditional distributions can be realized in dynamical systems when considering a suitable notion of controllability and how we can use them for causal inference.

### 3.1 Problem Setting

The system description from equation 1 is stochastic and, hence, induces a probability distribution $\mathbb{P}(x)$ over trajectories $x$. Based on this, we define non-causality adapting standard notation from the do-calculus (Pearl, 1995) to dynamical systems as in equation 1:

**Definition 1** (Global non-causality). *The state variable $x_j$ does not cause $x_i$ if $\mathbb{P}(x_i \mid \mathrm{do}(x_j(0) = x_j^{\mathrm{I}}(0))) = \mathbb{P}(x_i \mid \mathrm{do}(x_j(0) = x_j^{\mathrm{II}}(0)))$ for all $x_j^{\mathrm{I}}(0)$ and $x_j^{\mathrm{II}}(0)$. The superscripts* I *and* II *denote two experimental designs, i.e., different choices of $x_j(0)$. Similarly, $u_j$ does not cause $x_i$ if $\mathbb{P}(x_i \mid \mathrm{do}(u_j = u_j^{\mathrm{I}})) = \mathbb{P}(x_i \mid \mathrm{do}(u_j = u_j^{\mathrm{II}}))$ for all $u_j^{\mathrm{I}}$ and $u_j^{\mathrm{II}}$.*

The main objective of this article is to develop an algorithm to test for non-causality in the sense of definition 1 with theoretical guarantees. That is, we seek to test whether the initial condition of $x_j$ respectively the input trajectory $u_j$ changes the probability distribution of the resulting $x_i$ trajectory. The *do*-operator denotes that we force $x_j$ to a specific initial condition and $u_j$ to be a specific input trajectory, while all other initial conditions and input trajectories remain fixed between experiments. While this is a common assumption of the do-calculus, in dynamical systems, we cannot just *set* state variables to specific values. Instead, we need to consider a proper notion of *controllability*. In the following, we discuss how the do-operator can be applied to dynamical systems, i.e., how we can, based on an appropriate controllability notion, *steer* state variables toward specific values as required to test for non-causality. Subsequently, we develop an algorithm that identifies for each pair $x_i$ and $x_j$ and for each pair $x_i$ and $u_j$ of a control system as in equation 1, whether or not both variables are non-causal according to definition 1.

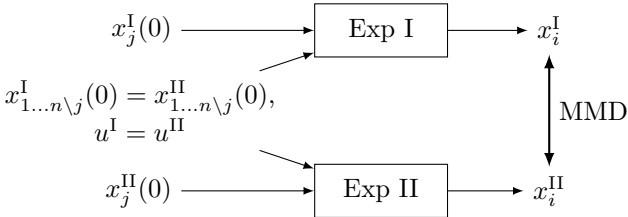

Figure 1: Experimental design for causal inference. *We design two experiments, where all initial conditions and input trajectories are constant except for $x_j(0)$. If the resulting trajectories of $x_i$ are different in both experiments, we have evidence that the change in $x_j(0)$ caused this change.*

### 3.2 Main Idea

For causal inference, we will exploit that we can influence equation 1 through $u$. To make the notion of *how* we can influence the system precise, we adopt controllability definitions for stochastic systems (Sunahara et al., 1974; Bashirov & Kerimov, 1997):

**Definition 2.** *The system described by equation 1 is said to be* completely $\epsilon$-controllable *in probability $\eta$ in the normed square sense in the time interval $[0, t_\mathrm{f}]$ if for all desired states $x_\mathrm{des}$ and initial states $x(0)$ from $\mathcal{X}$, there exists an input sequence $u$ from $\mathcal{U}$ such that $\Pr\{\|x(t_\mathrm{f}) - x_\mathrm{des}\|_2^2 \geq \epsilon\} \leq 1 - \eta$, with $0 < \eta < 1$.*

A variety of methods exist to identify or learn models for systems as in equation 1, e.g., Gaussian process regression (Williams & Rasmussen, 2006) or fitting linear state space models using least squares (Ljung, 1999). In the following, we assume that we can obtain an estimate $\hat{f}$ of the investigated system in equation 1 (including an estimate of the distribution of $v(t)$) using techniques from system identification and model learning. The concrete method that is used to obtain $\hat{f}$ is independent of the developed causal identification procedure. However, in later sections, we will specify requirements the model estimate $\hat{f}$ needs to satisfy. This model estimate $\hat{f}$, obtained without further assumptions or physical insights, will, due to the stochasticity of equation 1, almost surely entail spurious correlation and suggest causal influences that are actually not present in the physical system. Nevertheless, it will allow us to (approximately) steer the system to specific initial conditions and start experiments from there.

We propose two types of experiments to test for causal relations. For the first type, we investigate whether $x_j$ causes $x_i$. We conduct two experiments (denoted by I and II) with different initial conditions $x_j^\mathrm{I}(0) \neq x_j^\mathrm{II}(0)$, while all others are kept the same (cf. figure 1). This can be formalized as

$$x_\ell^\mathrm{I}(0) = x_\ell^\mathrm{II}(0) \text{ for all } \ell \neq j, \quad x_j^\mathrm{I}(0) \neq x_j^\mathrm{II}(0), \quad u_\ell^\mathrm{I}(t) = u_\ell^\mathrm{II}(t) \text{ for all } \ell, t. \tag{2a}$$

By comparing the resulting trajectories of $x_i^\mathrm{I}$ and $x_i^\mathrm{II}$, we can check whether the change in $x_j(0)$ caused a different behavior. For checking the similarity of trajectories, we will use the MMD, whose mathematical definition we provide in the next section. The second type of experiments is analogous to the first, but instead of varying initial conditions, we consider different input trajectories $u_j^\mathrm{I} \neq u_j^\mathrm{II}$,

$$x_\ell^\mathrm{I}(0) = x_\ell^\mathrm{II}(0) \text{ for all } \ell, \quad u_\ell^\mathrm{I}(t) = u_\ell^\mathrm{II}(t) \text{ for all } \ell \neq j, \text{ for all } t, \quad u_j^\mathrm{I}(t) \neq u_j^\mathrm{II}(t) \text{ for all } t. \tag{2b}$$

**Remark 1.** *Note that for the testing experiments, we design open-loop trajectories. That is, the input cannot depend on the system's current state. This is essential since it creates independent trajectories for which we can leverage the MMD as a similarity measure.*

### 3.3 Road-map

We propose an algorithm that consists of two steps: *(i)* we design experiments, and *(ii)* we analyze the resulting data using the MMD. The data obtained from these experiments must fulfill specific requirements to allow for proper causal inference. In the next section, we provide the mathematical definition of the MMD and deduce the requirements on the experimental data. We then subsequently develop a suitable

experimental design based on the controllability notion stated in definition 2. We state conditions under which this experimental design yields data from which the MMD can provably infer the true causal structure in the infinite sample limit and derive a hypothesis test for finitely many samples. Until here, we focus on a rigorous convergence analysis. However, this requires us to do many experiments on the real system. In section 5, we propose a heuristic algorithm that is more efficient in terms of the number of experiments and computations but forgoes some of the guarantees. Lastly, in section 6, we demonstrate the method's applicability on a real robotic system and a quadruple tank process and present comparisons with a sparse identification and a causal discovery method on a synthetic linear example.

## 4 Causal Identification for Dynamical Systems

We will now develop the causality testing procedure. First, we introduce the MMD (Gretton et al., 2012), which we shall use as a similarity measure. The MMD can be used to check whether two probability distributions $\mathbb{P}$ and $\mathbb{Q}$ are equal based on samples drawn from these distributions. Let $X$ and $Y$ be samples drawn i.i.d. from $\mathbb{P}$ and $\mathbb{Q}$, respectively. Further, let $\mathcal{H}$ be a *reproducing kernel Hilbert space* (Sriperumbudur et al., 2010), with canonical feature map $\phi \colon \mathcal{X} \to \mathcal{H}$. The MMD is defined as

$$\mathrm{MMD}(\mathbb{P}, \mathbb{Q}) = \|\mathbb{E}_{X \sim \mathbb{P}}[\phi(X)] - \mathbb{E}_{Y \sim \mathbb{Q}}[\phi(Y)]\|_{\mathcal{H}}. \tag{3}$$

The feature map $\phi$ can be expressed in terms of a kernel function $k(\cdot, \cdot)$, where $k(x, y) = \langle \phi(x), \phi(y) \rangle_{\mathcal{H}}$. If the kernel is characteristic, we have $\mathrm{MMD}(\mathbb{P}, \mathbb{Q}) = 0$ if, and only if, $\mathbb{P} = \mathbb{Q}$ (Fukumizu et al., 2008; Sriperumbudur et al., 2011). In the remainder of the paper, we always assume a characteristic kernel (e.g., the Gaussian kernel).

**Remark 2.** *In general, also other measures that compare probability distributions may be used for our algorithm. An overview of such methods is provided in Sriperumbudur et al. (2012). A popular example would be the Kullback-Leibler divergence (Kullback & Leibler, 1951). In this paper, we propose to use the MMD since it allows us to compare probability distributions without actually estimating them, provides theoretical guarantees, and can be computed efficiently.*

In the following, we derive conditions that allow one to provably identify causal relations. We investigate three settings. First, we discuss the case where we can precisely steer the system to desired initial conditions (i.e., $\epsilon = 0$ in definition 2). We then extend this to $\epsilon > 0$, which requires a stricter controllability definition. Finally, we show that for linear systems with additive Gaussian noise, a widely studied class of systems, the conditions stated by Kalman (Kalman, 1960a) are sufficient, and the identification is substantially easier.

### 4.1 Exact Controllability

When considering control systems, instead of obtaining single i.i.d. samples from stationary distributions, we receive sequences of random variables sampled from a stochastic process as in equation 1. This data is often non-i.i.d.. The objects of interest, whose distributions we want to compare, are then the $x_i$ trajectories obtained from two different experimental settings. To simplify notation, we denote a trajectory obtained from the first setting by $x_i^{\mathrm{I}}$ and the joint probability distribution over the trajectory states by $\mathbb{P}^{\mathrm{I}}$, and equivalently for $x_i^{\mathrm{II}}$ and $\mathbb{P}^{\mathrm{II}}$. We sample from $\mathbb{P}^{\mathrm{I}}$ and $\mathbb{P}^{\mathrm{II}}$ by designing two experiments as in equation 2 with fixed length $T$ and repeating each experiment $m$ times. That is, we obtain $2m$ sequences of $T$ random variables sampled at discrete intervals of fixed length (i.e., $t \to t+1$ always has the same length). These samples we denote by $\mathbf{x}_i^{\mathrm{I}}, \mathbf{x}_i^{\mathrm{II}} \in \mathbb{R}^{m \times T}$. Note that *(i)* all $m$ sequences are sampled from the same distributions $\mathbb{P}^{\mathrm{I}}$ and $\mathbb{P}^{\mathrm{II}}$, *(ii)* while the distributions are non-stationary along time, the distributions for multiple experiments of fixed length $T$ are identical, and *(iii)* all $m$ sequences are independent of each other. The MMD then reads

$$\mathrm{MMD}(\mathbf{x}_i^{\mathrm{I}}, \mathbf{x}_i^{\mathrm{II}}) = \|\mathbb{E}_{\mathbf{x}_i^{\mathrm{I}} \sim \mathbb{P}^{\mathrm{I}}}[\phi(\mathbf{x}_i^{\mathrm{I}})] - \mathbb{E}_{\mathbf{x}_i^{\mathrm{II}} \sim \mathbb{P}^{\mathrm{II}}}[\phi(\mathbf{x}_i^{\mathrm{II}})]\|_{\mathcal{H}}. \tag{4}$$

If we now design experiments using equation 2, we can check the similarity of $x_i$ trajectories with equation 4. Following Szabó & Sriperumbudur (2018), if we use a characteristic kernel in equation 4, the general properties of the MMD test are still valid given that the initial conditions $x^{\mathrm{I}}(0)$ and $x^{\mathrm{II}}(0)$, respectively the input trajectories $u^{\mathrm{I}}$ and $u^{\mathrm{II}}$, are i.i.d., which is the case for our design. That is, MMD $> 0$ suggests that the

distributions of the trajectories are different, so we can conclude that there is a causal influence. However, the other direction is less straightforward: for a system as in equation 1, the MMD may be zero, even though variables are dependent, as can be seen in the following example:

**Example 1.** *Assume a control system with $x_1(t+1) = x_1(t)x_2(t)$ and $x_2(t+1) = u(t)$, and an input signal $u(t)$ that is different from 0. If we choose $x_1(0) = 0$, the trajectory of $x_1$ will, despite the fact that $x_2$ clearly has a causal influence on $x_1$, always be 0 no matter the initial condition $x_2(0)$.*

To address this, we define the concept of local non-causality:

**Definition 3** (Local non-causality). *Let $\mathcal{X}_{nc} \subset \mathcal{X}$ and $\mathcal{U}_{nc} \subset \mathcal{U}$ with $\mathcal{X}_{nc} \cup \mathcal{U}_{nc} \neq \varnothing$. The state variable $x_j$ does* locally not cause *$x_i$ if $\mathbb{P}(x_i \mid \mathrm{do}(x_j(0) = x_j^{\mathrm{I}}(0))) = \mathbb{P}(x_i \mid \mathrm{do}(x_j(0) = x_j^{\mathrm{II}}(0)))$ for all $x_j^{\mathrm{I}}(0)$ and $x_j^{\mathrm{II}}(0)$ given that the sequence $x$ is entirely in $\mathcal{X}_{nc}$ and the sequence $u$ is entirely in $\mathcal{U}_{nc}$.[1] Similarly, the input variable $u_j$ does* locally not cause *$x_i$ if $\mathbb{P}(x_i \mid \mathrm{do}(u_j = u_j^{\mathrm{I}})) = \mathbb{P}(x_i \mid \mathrm{do}(u_j = u_j^{\mathrm{II}}))$ for all $u_j^{\mathrm{I}}$ and $u_j^{\mathrm{II}}$ given that $x$ is in $\mathcal{X}_{nc}$ and $u$ in $\mathcal{U}_{nc}$.*

The non-causality becomes global if $\mathcal{X}_{nc} = \mathcal{X}$ and $\mathcal{U}_{nc} = \mathcal{U}$. To properly test for causal relations, we need to ensure that the experimental design in equation 2 yields initial conditions and input trajectories that are not inside $\mathcal{X}_{nc}$ and $\mathcal{U}_{nc}$. For this, we propose to design experiments based on the estimated model $\hat{f}$. In particular, we utilize simulated trajectories based on the model $\hat{f}$, which we denote by $(\hat{x} \mid \hat{f})$ and where for each $t$ we have $\hat{x}(t) = \hat{f}(\hat{x}(0), u, \hat{v})$. We then need to make the following assumption about the system described by equation 1 and the estimated model $\hat{f}$:

**Assumption 1.** *Consider a dynamical system as in equation 1 for which $\mathcal{X}_{nc} \cup \mathcal{U}_{nc} \neq \varnothing$. Consider further two independent experimental designs following equation 2 with initial conditions $x^{\mathrm{I}}(0), x^{\mathrm{II}}(0)$ and input trajectories $u^{\mathrm{I}}, u^{\mathrm{II}}$ for the first, and initial conditions $x^{\mathrm{III}}(0), x^{\mathrm{IV}}(0)$ and input trajectories $u^{\mathrm{III}}, u^{\mathrm{III}}$ for the second experiment. Assume that all individual inputs $u(t)$ that are part of $u^{\mathrm{III}}$ or $u^{\mathrm{IV}}$ are in $\mathcal{U}_{nc}$ and that all states $x(t)$ that are part of the simulated trajectories $(\hat{x}^{\mathrm{III}} \mid \hat{f})$ or $(\hat{x}^{\mathrm{IV}} \mid \hat{f})$ are inside $\mathcal{X}_{nc}$. Further, assume that for the first experiment there exists some $u(t)$ or some $x(t)$ that is not part of $\mathcal{U}_{nc}$ respectively $\mathcal{X}_{nc}$. For such cases, we assume $\mathrm{MMD}(\hat{x}_i^{\mathrm{III}}, \hat{x}_i^{\mathrm{IV}} \mid \hat{f}) < \mathrm{MMD}(\hat{x}_i^{\mathrm{I}}, \hat{x}_i^{\mathrm{II}} \mid \hat{f})$.*

This assumption *does not* require that $\hat{f}$ captures the causal structure. That is, if variable $x_j$ does not cause variable $x_i$, the model $\hat{f}$ may still suggest that they are causally related—such modeling errors will then be accounted for through the identification procedure we propose in this paper. However, we require that $\hat{f}$ captures that the influence of $x_j$ on $x_i$ is lower in regions in which they are locally non-causal. Intuitively, the assumption says that if $x_i$ does not influence $x_j$ in certain parts of the state/action space, our model may, due to spurious correlation, still assign some influence of $x_i$ on $x_j$ in those regions. Nevertheless, we expect the model to assign a stronger influence in regions where the influence stems not only from spurious correlations but also from an actual physical influence of $x_i$ on $x_j$. Given a reasonable choice of system identification or model learning technique and a sufficiently rich excitation signal, this assumption will typically be satisfied in practice, as shown in the example below. It further follows that simulated trajectories in regions of local non-causality have a smaller MMD than trajectories in other regions.

**Example 1** (cont). *Given assumption 1, if we simulate experiments using equation 2a and compute the MMD for the resulting $\hat{x}_1$, the MMD will be* lower *if we choose $\hat{x}_1(0) = 0$ than for any other choice of $\hat{x}_1(0)$. To validate assumption 1, we identify the system using a regularization based nonlinear system identification algorithm called sparse identification of nonlinear dynamics (SINDy) (Brunton et al., 2016). By exciting the system for 500 time steps and using the SINDy implementation from de Silva et al. (2020), we receive a model. Simulating the MMD for different initial conditions $x_1(0)$ for $x_2^{\mathrm{I}}(0) = -10$ and $x_2^{\mathrm{II}}(0) = 10$ then reveals that the MMD has a minimum at $x_1(0) = 0$ (cf. figure 2), i.e., the identified model satisfies assumption 1. We will introduce the finite sample approximation for the MMD used in figure 2 in section 4.4 and we provide a further example to empirically verify assumption 1 for a system with a hysteresis in appendix B.*

We now specify the experimental design. To avoid regions of local non-causality, we propose to maximize the MMD given the model estimate. Thus, for checking whether $x_j$ causes $x_i$, we choose input trajectories

---

[1] In general, there may exist different $\mathcal{X}_{nc}^{ij}$ for each combination of $x_i$ and $x_j$ (and likewise for $\mathcal{U}_{nc}$). We can also cover this case. However, we omit it here to simplify notation.

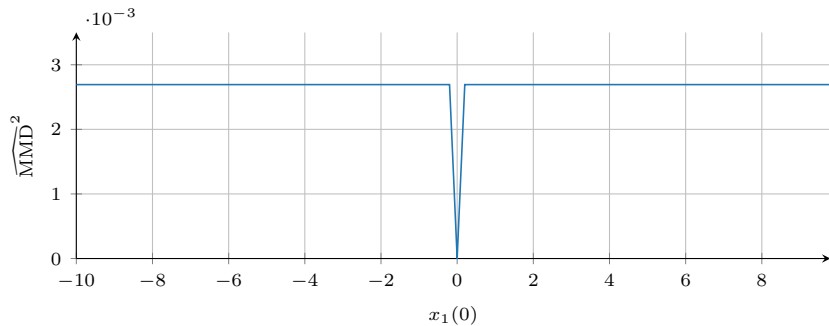

Figure 2: MMD of simulated experiments with different initial conditions $x_1(0)$ for the system from example 1. *The MMD has a minimum at $x_1(0) = 0$, i.e., it reflects the local non-causality. The finite sample approximation $\widehat{\mathrm{MMD}}^2$ of the MMD used in this figure will be introduced in section 4.4.*

of length $T$ and initial conditions that solve

$$\max_{x^{\mathrm{I}}(0), x^{\mathrm{II}}(0), u^{\mathrm{I}}, u^{\mathrm{II}}} \mathrm{MMD}(\hat{x}_i^{\mathrm{I}}, \hat{x}_i^{\mathrm{II}} \mid \hat{f})$$

$$\text{subject to } x_\ell^{\mathrm{I}}(0) = x_\ell^{\mathrm{II}}(0) \text{ for all } \ell \neq j \quad u_\ell^{\mathrm{I}}(t) = u_\ell^{\mathrm{II}}(t) \text{ for all } \ell, t. \tag{5a}$$

We will discuss how to handle this optimization problem in practice in section 5.2. In particular, in practice, we typically do not require a global solution to equation 5a. If we want to check whether $u_j$ causes $x_i$, we choose input trajectories and initial conditions by solving

$$\max_{x^{\mathrm{I}}(0), x^{\mathrm{II}}(0), u^{\mathrm{I}}, u^{\mathrm{II}}} \mathrm{MMD}(\hat{x}_i^{\mathrm{I}}, \hat{x}_i^{\mathrm{II}} \mid \hat{f})$$

$$\text{subject to } x_\ell^{\mathrm{I}}(0) = x_\ell^{\mathrm{II}}(0) \text{ for all } \ell \quad u_\ell^{\mathrm{I}}(t) = u_\ell^{\mathrm{II}}(t) \text{ for all } \ell \neq j, t. \tag{5b}$$

Before stating our main theorem, we need one further assumption. Given the system model from equation 1, we could also have systems for which the influence of $x_j$ on $x_i$ becomes only apparent after a certain number of time steps. To guarantee that we identify the true causal structure, we need to assume that this delay is at most equal to the length of an experiment $T$.

**Assumption 2.** *Consider a pair $(x_i, x_j)$ for which $x_j$ has a causal influence on $x_i$ as per definition 3. We assume that changes of $x_j$, when outside of regions of local non-causality, cause a change in $x_i$ after at most $T$ discrete time steps.*

We can now state the main theorem:

**Theorem 1.** *Consider a completely $\epsilon$-controllable system described by equation 1 with $\epsilon = 0$ that fulfills assumptions 1 and 2. Let experiments be designed according to equation 5 for a fixed experiment length $T$ and repeated infinitely often (i.e., we have $\boldsymbol{x}_i^{\mathrm{I}}, \boldsymbol{x}_i^{\mathrm{II}} \in \mathbb{R}^{m \times T}$ with $m \to \infty$). Then: $\mathrm{MMD}(\boldsymbol{x}_i^{\mathrm{I}}, \boldsymbol{x}_i^{\mathrm{II}}) = 0$ if, and only if, $x_j$ respectively $u_j$ does not cause $x_i$ as per definition 1.*

*Proof.* Let variables be non-causal. Then, we have by definition 1, $\mathbb{P}(x_i \mid \mathrm{do}(x_j(0) = x_j^{\mathrm{I}}(0))) = \mathbb{P}(x_i \mid \mathrm{do}(x_j(0) = x_j^{\mathrm{II}}(0)))$. That is, the distribution of $x_i$ in both experiments is equal. Thus, $\mathrm{MMD}(\mathbf{x}_i^{\mathrm{I}}, \mathbf{x}_i^{\mathrm{II}}) = 0$ follows from Gretton et al. (2012). Now, assume $\mathrm{MMD}(\mathbf{x}_i^{\mathrm{I}}, \mathbf{x}_i^{\mathrm{II}}) = 0$. This implies that the distribution of $x_i$ is equal in both experiments (Gretton et al., 2012), i.e., $\mathbb{P}(x_i \mid \mathrm{do}(x_j(0) = x_j^{\mathrm{I}}(0))) = \mathbb{P}(x_i \mid \mathrm{do}(x_j(0) = x_j^{\mathrm{II}}(0)))$. This could be the case because *(i)* $x_i$ and $x_j$ are non-causal or *(ii)* we have that $x \in \mathcal{X}_{\mathrm{nc}}$ and $u \in \mathcal{U}_{\mathrm{nc}}$. Due to assumption 1 and equation 5, there exists a $t$ for which either $x \notin \mathcal{X}_{\mathrm{nc}}$ or $u \notin \mathcal{U}_{\mathrm{nc}}$. Thus, we are in case *(i)* and $x_j$ does not cause $x_i$. The proof for $u_j$ and $x_i$ follows analogously. $\square$

**Remark 3.** *Assumption 1 ensures that, if we design experiments using the optimization algorithm in equation 5, we are not inside a region of local non-causality. However, even if we weaken the assumption, the claim of theorem 1 would still hold. Since equation 5 finds the experiment design that maximizes the MMD,*

*the algorithm only fails if our model estimate assumes the influence of one variable on the other to be strongest in regions where there is actually no causal influence. Given a reasonable model class and excitation signal for the initial model estimation, this should not occur in practice. We still keep assumption 1 in its stronger form as it allows us to also make causal inference in practical settings where we may not be able to globally solve equation 5.*

**Remark 4.** *Instead of doing multiple experiments of fixed length, one might consider doing one long experiment. This design has several disadvantages. First, if the process is stable, the probability distribution can become independent of the initial conditions over time. Then, causal influences are only visible during the transient. Second, if we deal with non-ergodic systems, where time and spatial average are not the same, we only have a valid testing procedure if we do multiple runs of each experiment. Lastly, doing multiple experiments ensures that the different runs are independent of each other. Data generated by a single long experiment can be correlated, and obtaining a valid test is way more involved (Solowjow et al., 2020).*

## 4.2 $\epsilon$-Controllability

For a stochastic system as in equation 1, it is in general impossible to steer the system exactly to the initial conditions suggested by equation 5; i.e., we need to resort to controllability with $\epsilon > 0$ (cf. definition 2). Nevertheless, even in such cases, it is still possible to guarantee the consistency of the causality testing procedure. However, we need a stricter definition of controllability.

**Definition 4.** *Let the system given in equation 1 be $\epsilon$-controllable according to definition 2, and consider some arbitrary, but fixed $x_{\ell,\mathrm{des}}^*$. Then, the system given in equation 1 is said to be* completely $\epsilon$-controllable in distribution *if, for any $x(0)$ and any $x_{\mathrm{des}}$ with $x_{\ell,\mathrm{des}} = x_{\ell,\mathrm{des}}^*$, there exists an input sequence $u(0\!:\!t_{\mathrm{f}})$ such that $x_\ell(t_{\mathrm{f}})$ always follows the same distribution; i.e., $\mathbb{P}(x_\ell(t_{\mathrm{f}})) = \mathbb{P}^*$ for some $\mathbb{P}^*$ that does not depend on $x(0)$ or any component of $x_{\mathrm{des}}$ except $x_{\ell,\mathrm{des}} = x_{\ell,\mathrm{des}}^*$ and $\Pr\{\|x(t_{\mathrm{f}}) - x_{\mathrm{des}}\|_2^2 \geq \epsilon\} \leq 1 - \eta$, with $0 < \eta < 1$.*

In other words, the definition states that, for any $x(0)$, we can generate input trajectories that guarantee that the fixed component $x_{\ell,\mathrm{des}}^*$ of $x_{\mathrm{des}}$ is matched in distribution. Linear systems with additive Gaussian noise that are controllable following Kalman (1960a) are also controllable in the sense of definition 4, as we show in the appendix. We further need to make an assumption about the initial conditions suggested by equation 5. To steer the system to those initial conditions without ending up in a region of local non-causality, we need that states that are $\epsilon$-close to those initial conditions are not in a region of local non-causality.

**Assumption 3.** *For the initial conditions suggested by equation 5, we assume that there exist open balls with radius $\sqrt{\epsilon}$ centered around each element of $x^{\mathrm{I}}(0)$ and $x^{\mathrm{II}}(0)$ that are outside of the local non-causality sets.*

This assumption is not very strong since equation 5 suggests initial conditions for which the influence of $x_j$ respectively $u_j$ on $x_i$ is maximal. Thus, it is unlikely that these initial conditions are close to regions of local non-causality. The main reason why this assumption is needed is to exclude corner cases in which the causal influence only exists in single points of the state and input space. In such cases, the probability of successfully steering the system to those points is zero. However, considering the variables as non-causal may then anyway be reasonable. Equipped with these three assumptions, we can now state:

**Corollary 1.** *Consider a system as in equation 1 that is completely $\epsilon$-controllable in distribution according to definition 4 and fulfills assumptions 1, 2, and 3. Let experiments be designed as in equation 5 for a fixed experiment length $T$, trajectories that steer the system to the initial conditions of the experiments be chosen such that $P(x_\ell^{\mathrm{I}}(0)) = P(x_\ell^{\mathrm{II}}(0))$ for all $\ell \neq j$, and experiments be repeated infinitely often. Then: $\mathrm{MMD}(\boldsymbol{x}_i^{\mathrm{I}}, \boldsymbol{x}_i^{\mathrm{II}}) = 0$ if, and only if, $x_j$ respectively $u_j$ does not cause $x_i$ according to Definition 1.*

*Proof.* Let variables be non-causal. Then, we have $\mathbb{P}(\mathbf{x}_\ell^{\mathrm{I}}(0)) = \mathbb{P}(\mathbf{x}_\ell^{\mathrm{II}}(0))$ for all $\ell \neq j$, thus, also the distribution of the obtained $x_i$ trajectories is equal and we have $\mathrm{MMD}(\mathbf{x}_i^{\mathrm{I}}, \mathbf{x}_i^{\mathrm{II}}) = 0$ (Gretton et al., 2012). Now, assume $\mathrm{MMD}(\mathbf{x}_i^{\mathrm{I}}, \mathbf{x}_i^{\mathrm{II}}) = 0$. This implies that the distribution of $x_i$ is equal in both experiments (Gretton et al., 2012). By assumption 1, existing local non-causalities are reflected by the model and thus, equation 5 will suggest experiments outside of such regions. Assumption 3 ensures that we can steer the system to those regions. Thus, if distributions are equal, non-causality must be global as in definition 1. $\qquad\square$

### 4.3 Linear Systems

Local non-causality as in definition 3 is a nonlinear phenomenon. If we assume equation 1 to be linear time-invariant (LTI) with Gaussian noise $v(t)$, we can reveal the true causal structure without the optimization procedure in equation 5, making this case substantially easier. For an LTI system, equation 1 reads

$$x(t) = A^t x(0) + \sum_{i=0}^{t-1} A^i (Bu(t-1-i) + v(t-1-i)), \tag{6}$$

with state transition matrix $A \in \mathbb{R}^{n \times n}$, input matrix $B \in \mathbb{R}^{n \times m}$, and $v(t) \sim \mathcal{N}(0, \Sigma_{\mathrm{v}})$. The system in equation 6 is controllable as per definition 4 if it satisfies the classical controllability condition from Kalman (1960a), i.e., if the matrix $\begin{pmatrix} B & AB & \ldots & A^{n-1}B \end{pmatrix}$ has full row rank, as we show in lemma 3 in the appendix. We can then state the following theorem, whose proof is provided in the appendix:

**Theorem 2.** *Assume an LTI system as in equation 6, whose $A$ and $B$ matrices satisfy Kalman's controllability condition. Let experiments be designed as in equation 2a and equation 2b, respectively. Then:* $\mathrm{MMD}(\boldsymbol{x}_i^{\mathrm{I}}, \boldsymbol{x}_i^{\mathrm{II}}) = 0$ *if, and only if, $x_j$ respectively $u_j$ does not cause $x_i$ as per definition 1.*

### 4.4 Test with Finite Samples

Until here, we derived guarantees in the infinite sample limit. In practice, we can only carry out finitely many experiments, i.e., we have finitely many samples of the random variable sequence $x_i$. Thus, we need a finite sample approximation of the MMD.

**Lemma 1.** *Consider $m$ experiments with fixed length $T$, i.e., $\boldsymbol{x}_i \in \mathbb{R}^{m \times T}$ but now we also have $m < \infty$. An unbiased empirical estimate of the squared population MMD can be computed as*

$$\widehat{\mathrm{MMD}}^2(\boldsymbol{x}_i^{\mathrm{I}}, \boldsymbol{x}_i^{\mathrm{II}}) = \frac{1}{m(m-1)} \sum_{r \neq s}^m (k(^r x_i^{\mathrm{I}}, {}^s x_i^{\mathrm{II}}) + k(^r x_i^{\mathrm{II}}, {}^s x_i^{\mathrm{II}}) - k(^r x_i^{\mathrm{I}}, {}^s x_i^{\mathrm{II}}) - k(^s x_i^{\mathrm{I}}, {}^r x_i^{\mathrm{II}})), \tag{7}$$

*where $^r x_i^{\mathrm{I}}$ denotes element $r$ of the $x_i$ trajectories from experiment I.*

*Proof.* The $m$ trajectories for $x^{\mathrm{I}}$ and $x^{\mathrm{II}}$ are independent and follow $\mathbb{P}^{\mathrm{I}}$ and $\mathbb{P}^{\mathrm{II}}$, respectively. Thus, we are in the same setting as in (Gretton et al., 2012, lem. 6) and the proof follows as shown therein. $\square$

For a finite sample approximation, we, in general, have $\mathrm{MMD}^2(\mathbf{x}_i^{\mathrm{I}}, \mathbf{x}_i^{\mathrm{II}}) > 0$ even if $\mathbb{P}^{\mathrm{I}} = \mathbb{P}^{\mathrm{II}}$.[2] Thus, we need to do a hypothesis test and derive a test statistic. We assume the null hypothesis

$$H_0: \quad \mathbb{P}^{\mathrm{I}} = \mathbb{P}^{\mathrm{II}} \tag{8}$$

and obtain the test statistic from the following result.

**Lemma 2.** *Assume $0 \leq k(^r x_i^{\mathrm{I}}, {}^s x_i^{\mathrm{II}}) \leq K$. Then*

$$\Pr_{\boldsymbol{x}_i^{\mathrm{I}}, \boldsymbol{x}_i^{\mathrm{II}}} \{ \widehat{\mathrm{MMD}}^2(\boldsymbol{x}_i^{\mathrm{I}}, \boldsymbol{x}_i^{\mathrm{II}}) - \mathrm{MMD}^2(\mathbb{P}^{\mathrm{I}}, \mathbb{P}^{\mathrm{II}}) > \gamma \} \leq \exp\left( \frac{-\gamma^2 m_2}{8K^2} \right),$$

*where $m_2 := \lfloor m/2 \rfloor$. The hypothesis test of level $\alpha$ for the null hypothesis in equation 8 has the acceptance region*

$$\widehat{\mathrm{MMD}}^2(\boldsymbol{x}_i^{\mathrm{I}}, \boldsymbol{x}_i^{\mathrm{II}}) < \left( \frac{4K}{\sqrt{m}} \sqrt{\log(\alpha^{-1})} \right).$$

*Proof.* The $m$ trajectories for $x^{\mathrm{I}}$ and $x^{\mathrm{II}}$ are independent and follow $\mathbb{P}^{\mathrm{I}}$ and $\mathbb{P}^{\mathrm{II}}$, respectively. Thus, the proof follows from theorem 10 and corollary 11 in Gretton et al. (2012). $\square$

---

[2]Note that the unbiased estimate in lemma 1 can even become negative, cf. the discussion after lemma 6 in Gretton et al. (2012).

## 5 Implementation

The results in section 4 show that we are able to detect whether the variables $x_j$ or $u_j$ have a causal influence on $x_i$. In practical implementations, two challenges remain: first, we want to minimize the number of experiments we need to carry out on the physical platform. Second, we may be unable to obtain a global solution to equation 5.

### 5.1 Heuristic Test with Finite Samples

The test statistic provided in lemma 2 enjoys a theoretical foundation. However, the threshold decreases as $m^{-1/2}$, i.e., we need many experiments to not be overly conservative. While more efficient test statistics exist (e.g., achieved through subsampling as discussed in section 6 of Gretton et al. (2012)), generating all this data through experiments on real-world systems is often undesirable, e.g., because it is time-consuming and may cause excessive wear and tear on the hardware. Thus, we propose an alternative test statistic that can be obtained from the model estimate $\hat{f}$. This alternative test statistic is heuristic and forgoes the theoretical properties but is efficient to implement and yields good results in practice, as we show in section 6.

We estimate a model $\hat{f}_{i,\mathrm{nc}}$ that assumes $x_i$ and $x_j$ respectively $u_j$ are non-causal (i.e., we do not use the data $x_j$ respectively $u_j$ when estimating $\hat{f}_{i,\mathrm{nc}}$). We propose to use this model to decide whether to accept the null hypothesis of $x_i$ and $x_j$ respectively $u_j$ being non-causal. That is, we replace our current model $\hat{f}_i$ for state component $x_i$ with $\hat{f}_{i,\mathrm{nc}}$ if

$$\widehat{\mathrm{MMD}}^2(x_i^{\mathrm{I}}, x_i^{\mathrm{II}}) < \mathbb{E}[\widehat{\mathrm{MMD}}^2(\hat{x}_i^{\mathrm{I}}, \hat{x}_i^{\mathrm{II}} \mid \hat{f}_{i,\mathrm{nc}})] + \nu\sqrt{\mathrm{Var}[\widehat{\mathrm{MMD}}^2(\hat{x}_i^{\mathrm{I}}, \hat{x}_i^{\mathrm{II}} \mid \hat{f}_{i,\mathrm{nc}})]}. \tag{9}$$

Expected value and variance in equation 9 can be estimated through Monte Carlo simulations. For these simulations, we use the true initial conditions $x^{\mathrm{I}}(0)$ and $x^{\mathrm{II}}(0)$. That way, we account for uncertainty due to unequal initial conditions between experiments. The significance level of the test can be adjusted through $\nu$ using Chebyshev's inequality (Chebyshev, 1867).

### 5.2 Experimental Design

The framework for testing causality between state variables is summarized in algorithm 1 (and works analogously for inputs). After having obtained an initial model (see ll. 1-2 in algorithm 1), we run equation 5a to find initial conditions and input trajectories for testing non-causality of one specific combination of $x_\ell$ and $x_j$ (l. 6). The optimization in equation 5a may be arbitrarily complex or even intractable, depending on the chosen model class. However, finding a global optimum of equation 5a is not necessary. The goal of the optimization procedure is to avoid regions of local non-causality. We thus optimize equation 5a until it is above a threshold $\delta_1$ to be confident that we are not in a region of local non-causality. In practical applications, we can often already achieve this through a reasonable initialization of the optimization problem, i.e., by choosing initial conditions for $x_j$ as far apart as possible. We then run the designed experiment and collect the data (l. 7). Ideally, we would like to use data from this single experiment to test for the causal influence of $x_j$ on all other state components. Thus, we check for which $x_i$ the experiment yields an expected MMD that is higher than a second threshold $\delta_2$ and do the hypothesis test for all of those (ll. 9-13).

## 6 Evaluation

We evaluate the framework on three systems. First, we identify the causal structure of one arm of the robot Apollo (Kappler et al., 2018) shown in figure 3 in section 6.1. Then, we demonstrate the causal identification of a simulated quadruple tank process (cf. figure 5) in section 6.2. In both cases, we present the general setup and results in the following, while we defer implementation details to the appendix. Lastly, we discuss a synthetic linear toy example. We use this example to highlight once more the importance of considering a notion of controllability and to compare our method to a sparse identification and a causal discovery algorithm[3]. For all experiments, we use a Gaussian kernel to compute the MMD.

---

[3]Code for both simulation examples is available at `https://github.com/baumanndominik/identifying_causal_structure`.

---

**Algorithm 1** Pseudocode of the proposed framework.

---

1: Excite system with input signal, collect data
2: Obtain $\hat{f}$ through black-box system identification
3: **for** $x_j$ in $x$ **do**
4:     $x_\text{test} = [x_1, \ldots, x_n]$             ▷ states to be tested for non-causality
5:     **for** $x_\ell$ in $x_\text{test}$ **do**         ▷ design experiment to test whether $x_j$ causes $x_\ell$
6:         Run equation 5a until $\mathbb{E}[\widehat{\text{MMD}}^2(\hat{x}_\ell^\text{I}, \hat{x}_\ell^\text{II} \mid \hat{f})] > \delta_1$
7:         Run causal experiments, collect data
8:         **for** $x_i$ in $x_\text{test}$ **do**      ▷ The experimental designed for $x_j$ and $x_\ell$ might
                                                                  also yield a valid test for other $x$
9:             **if** $\mathbb{E}[\widehat{\text{MMD}}^2(\hat{x}_i^\text{I}, \hat{x}_i^\text{II} \mid \hat{f})] > \delta_2$ **then**   ▷ If the empirical MMD is above the threshold
                                                                  for some other $x_i$, also test that $x_i$
10:                 Obtain $\hat{f}_{i,\text{nc}}$
11:                 Obtain test statistic via Monte Carlo simulations
12:                 **if** equation 9 holds **then**        ▷ independence test
13:                     $\hat{f}_i = \hat{f}_{i,\text{nc}}$
14:                 Delete $x_i$ from $x_\text{test}$

---

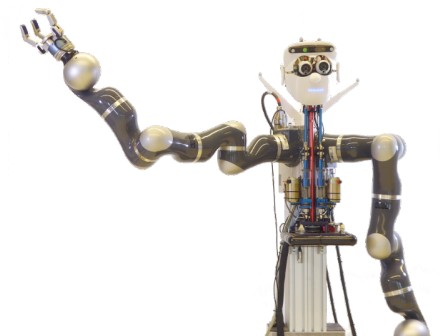 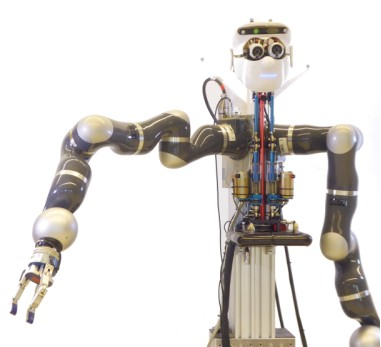

Figure 3: The robot showing initial postures for two experiments.

## 6.1 Robot Experiments

We consider kinematic control of the robot; that is, we can command desired angular velocities to the joints, which are then tracked by low-level controllers (taking care, among other things, of the robot dynamics) (Berenz et al., 2021). As measurements, we receive the joint angles. The goal of the causal identification is to learn which joints influence each other and which joints are influenced by which inputs. We consider four joints of the robot arm in the following experiments. From the robot arm's design, we know its kinematic structure, which is described by $\dot{\phi}_i = u_i$ for each joint, where $\phi_i$ denotes the angle of joint $i$ and $u_i$ the control input. That is, we expect each joint velocity $\dot{\phi}_i$ to depend only on the local input $u_i$ and not other variables. While the dynamics are approximately linear, we do not rely on this information and are thus in the setting discussed in section 4.2. In the following, we will investigate whether our proposed causal identification can automatically reveal this structure.

Following algorithm 1, we start by identifying a model $\hat{f}$. In this experiment, we estimate a linear state-space model. As expected, the initial model suggests that all joints are linked to each other and all inputs due to spurious correlations. We then design experiments for causality testing, example trajectories of such experiments are shown in figure 4 (left). The empirical squared MMD of the resulting trajectories is compared with the test statistic. The trajectories in figure 4 (left) already suggest that the experiments are in line with the kinematic model: while the two trajectories of joint 1 for different initial conditions of joint 3 are essentially equal (blue dashed and green dotted lines overlap), the trajectories of joint 3 for different choices of the third input are fundamentally different. This is also revealed through the proposed causality

test. The middle plots of figure 4 show the empirical squared MMD (left-hand side of equation 9) and the test threshold (right-hand side of equation 9) for the experiments that were conducted to test the influence of the initial conditions of the third joint (top) and of the third input signal (bottom) on all joints. As can be seen, the causal identification reveals that the third joint does not influence any other joint, and the third input only affects the third joint. Note that the third joint's trajectories are obviously different when choosing different initial conditions for the third joint. However, since this is expected, we subtract the initial condition in this case to investigate whether the movement starting from these distinct initial conditions differs. The remaining experiments (results are contained in the appendix) yield similar results. In summary, the causal identification successfully reveals the expected causal structure.

The experiment design in equation 5 requires us to solve an optimization problem. Nevertheless, algorithm 1 introduces two parameters, $\delta_1$ and $\delta_2$. These can be used to stop the optimization early in case the predicted MMD is high enough for us to be confident that we are not in a region of local non-causality ($\delta_1$) and that we can use the design to test for all influences of a joint or input signal ($\delta_2$). To design $\delta_1$ and $\delta_2$, we can look at the system's noise level and choose them some orders of magnitude higher. As discussed in section 5, a high predicted MMD can often already be achieved through sophisticated guesses, i.e., by choosing initial conditions far apart from each other and diverse input signals. We follow this approach. When testing for the influence of the third joint on others and choosing initial conditions that are far apart, we predict MMDs of around 0.5. The model we estimate initially for the robot arm has a noise standard deviation below $1 \times 10^{-4}$. That is, the predicted MMD is way above the noise level of the system and way above the MMDs we find in experiments. Thus, for any choice of $\delta_1$ and $\delta_2$ below 0.5, we can confidently accept this experiment design. If we were even more conservative and chose them above 0.5, we would need to optimize the experiment design further.

To investigate the generalization capability, we compare predictions of the model $\hat{f}_{\text{init}}$ obtained from the initial system identification and the model $\hat{f}_{\text{caus}}$ that was learned after revealing the causal structure. We use the same training data to estimate the model parameters in both cases. However, for $\hat{f}_{\text{caus}}$, we leverage the obtained knowledge of the causal structure when estimating parameters. In contrast, for $\hat{f}_{\text{init}}$ we do not take any prior knowledge into account. As test data, we use an experiment that was conducted to investigate the influence of the initial condition of joint 3 on the other joints and let both models predict the trajectory of joint 1. For this experiment, the initial angle of joint 3 is close to its maximum value, a case that is not contained in the training data. As can be seen in figure 4 (right), the predictions of $\hat{f}_{\text{caus}}$ (blue) are very close to the true data (green, dashed), i.e., the model can generalize well, while the predictions of $\hat{f}_{\text{init}}$ deviate significantly.

### 6.2 Quadruple Tank Process

The experimental demonstration in the previous section showed that the presented algorithm can successfully identify a real-world robotic system's causal structure. However, the causal structure is relatively straightforward, and dynamics are approximately linear. To stress the method's ability to perform similarly well on more complex structures and with nonlinear dynamics, we now consider the quadruple tank system from Johansson (2000), illustrated in figure 5. Its continuous-time dynamics are given by

$$
\begin{aligned}
\dot{x}_1 &= -\frac{a_1}{A_1}\sqrt{2gx_1} + \frac{a_3}{A_1}\sqrt{2gx_3} + \frac{\zeta_1}{A_1}u_1 \\
\dot{x}_2 &= -\frac{a_2}{A_2}\sqrt{2gx_2} + \frac{a_3}{A_2}\sqrt{2gx_4} + \frac{\zeta_2}{A_2}u_2 \\
\dot{x}_3 &= -\frac{a_3}{A_3}\sqrt{2gx_3} + \frac{1-\zeta_2}{A_3}u_2 \\
\dot{x}_4 &= -\frac{a_4}{A_4}\sqrt{2gx_4} + \frac{1-\zeta_1}{A_4}u_1,
\end{aligned}
\tag{10}
$$

where $x_i$ denotes the water level of tank $i$, $u_i$ the flow rate of pump $i$, $g$ the gravitational constant ($9.81\,\text{m}\,\text{s}^{-2}$), and the remaining constants as in figure 5. The dynamics of the quadruple tank process are, thus, clearly nonlinear. That is, we cannot expect good performance if we approximate them using a linear state-space

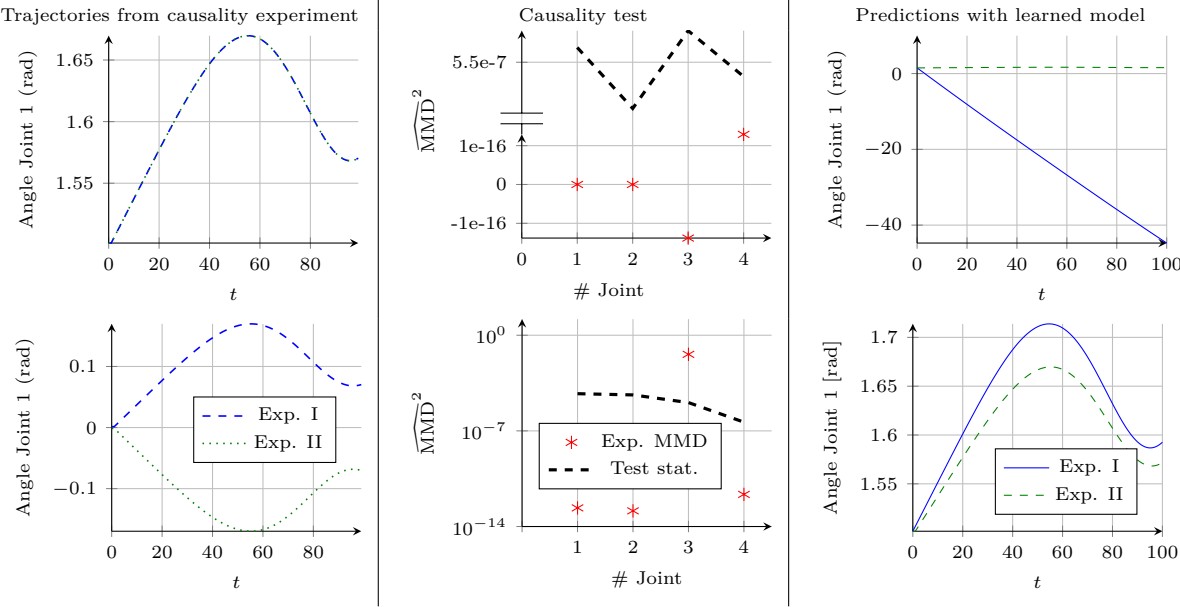

Figure 4: Causality tests and model evaluation for the robotic system. *Plots on the left show example trajectories of two experiments, in the middle the experimental MMD and the test threshold for joint 3, and on the right predictions based on the initial model and on the refined model after the causal identification.*

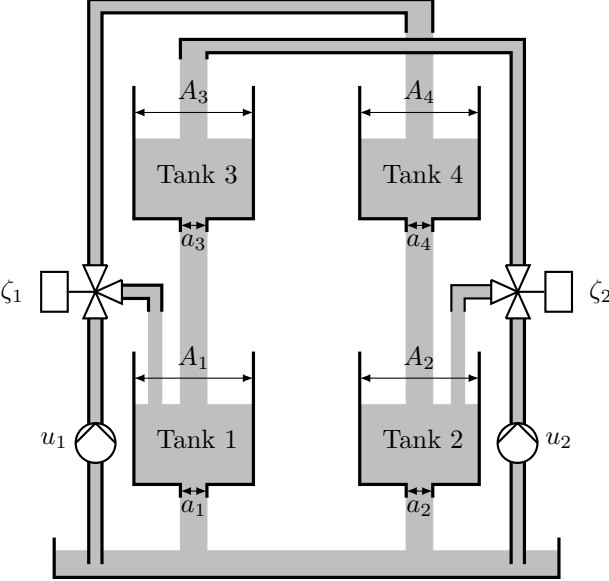

Figure 5: Schematic of the quadruple tank process from Johansson (2000).

model. Therefore, we discretize the system and use Gaussian processes (GPs) to model the dynamics, see Williams & Rasmussen (2006) for an introduction and details. In particular, we model each state of the system with a GP, where each GP uses all states and inputs to predict its assigned state variable at the next time step.

We proceed similarly as for the robot experiments. We select initial conditions through sophisticated guesses, i.e., far apart from each other. In a simulation, we can set initial conditions and do not need to steer the system there. However, to be more realistic, we sample initial conditions from a normal distribution

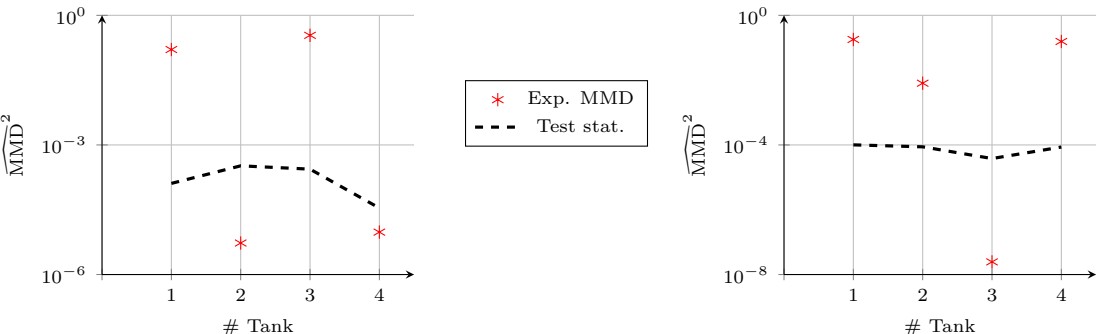

Figure 6: Causality tests for the quadruple tank system. *The plots show the results of the experiments testing the influence of the third tank on all others (left) and the influence of $u_1$ on all tanks (right).*

with mean equal to the selected initial conditions and variance of $1 \times 10^{-2}$. Thus, we only consider $\epsilon$-controllability.

Results of the testing procedure are shown in figure 6. The left plot shows test statistic and experimental MMD for the third tank's influence on all others. It reveals that the third tank influences itself and the first tank, which is in line with the schematic in figure 5 and the dynamics in equation 10. The right plot illustrates the test statistic and experimental MMD for the influence of $u_1$ on the tanks. Also these results are in accordance with figure 5, as the experiments suggest that $u_1$ influences all but the third tank. The remaining results are collected in section D in the appendix. Similarly as the ones in figure 6, they reveal the causal structure that can be inferred from figure 5.

In this case, the causal identification results in a reduction of computational complexity. Especially for the third and the fourth tanks, which are only influenced by $u_2$ respectively $u_1$ and by themselves, the input dimension of their corresponding GPs decreases from 10 to 3. The complexity of standard GP regression grows with $\mathcal{O}(n^3)$ with the number of datapoints $n$. Thus, if we can reduce the dimensions that need to be considered and, with that, the number of data points that we use for the regression by $70\,\%$ as in this example, we can considerably reduce the computational complexity.

### 6.3   Synthetic Example and Comparison

Lastly, we present a synthetic, linear example. We consider an LTI system as in equation 6, with

$$A = \begin{pmatrix} 0.9 & -0.75 & 1.2 \\ 0 & 0.9 & -1.1 \\ 0 & 0 & 0.7 \end{pmatrix} \quad B = \begin{pmatrix} 0.03 & 0 & 0 \\ 0 & 0.06 & 0 \\ 0.07 & 0 & 0.05 \end{pmatrix} \tag{11}$$

and Gaussian noise with standard deviation $1 \times 10^{-4}$. For this example, we apply algorithm 1 without the need for the optimization procedure since the example is linear. Again, we want to stress the importance of an appropriate notion of controllability. That is, instead of assuming that we can set the system to initial conditions, we always *steer* the system to the initial conditions required for that experiment. For this, we employ an approach to set-point tracking that has, for instance, been discussed in Pannocchia et al. (2005). Given a desired state $x_{\mathrm{des}}$, we seek a feedback control law of the form $u = Mx_{\mathrm{des}} + Fx$, i.e., a control law that depends both on the desired state and the current state. We obtain the gain matrix $F$ using standard methods from linear optimal control (Anderson & Moore, 2007), in particular, the linear quadratic regulator (LQR). Thus, we can rewrite the incremental dynamics of the system as

$$x(t+1) = \tilde{A}x(t) + BMx_{\mathrm{des}} + v(t), \tag{12}$$

where $\tilde{A} := A + BF$. We now choose the feedforward term $M$ such that the reference is matched in stationarity, i.e., we want to achieve

$$x = (I - \tilde{A})^{-1}BMx_{\mathrm{des}}. \tag{13}$$

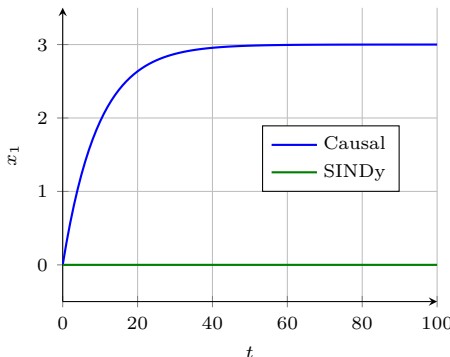

Figure 7: Comparison of prediction capabilities. *In blue predictions of the true model obtained after causal identification, in green the model obtained from SINDy with parameter choices from Brunton et al. (2016). We do not compare to PCMCI in the numerical experiment since PCMCI only reveals which causal influences exist, but not how strong they are, i.e., different from our algorithm and SINDy, it does not return a full system model that can be used in simulations.*

Thus, we need

$$M = ((I - \tilde{A})^{-1}B)^{-1} \tag{14}$$

to track the reference point. To compute $M$, we use the matrices $\hat{A}$ and $\hat{B}$ of the estimated model $\hat{f}$. We start the experiment once $\|x(t) - x_{\text{des}}\|_2 < 0.01$.

Analyzing the resulting data with the MMD lets us, similar as before, infer the true causal structure. In particular, the causal analysis reveals that $x_1$ does not cause $x_2$ nor $x_3$, $x_2$ does not cause $x_3$, and $u_2$ does not cause $x_3$.

In addition to those results, we compare our method with a sparse identification method and a causal discovery algorithm for this linear example. In both cases, we excite the system for 100 000 time-steps with a chirp as an input signal.

First, we use SINDy, as in example 1, to identify the underlying discrete-time system[4]. Sparse identification algorithms seek to find a trade-off between model complexity and goodness of fit. Thus, for such algorithms, a parameter needs to be selected to indicate how important accuracy is compared to complexity. When applying SINDy to the synthetic system, its success in finding the true causal structure depends on the choice of this parameter. We start with the parameter settings that were used for a three-dimensional linear example in Brunton et al. (2016). For those settings, the algorithm does not recognize the influence of $u_1$ on $x_1$, i.e., the first row of the $B$ matrix consists of zeros. In figure 7, we show the effects that this error can have by comparing predictions of the true model obtained after the causal identification with predictions of the model obtained from SINDy. For this, we set $u_1 = 10$, $u_2 = 0$, and $u_3 = -14$ and simulate both models for 100 time steps. As the SINDy model does not reflect the influence of $u_1$ on $x_1$, it assumes that $x_1$ does not move. However, the model obtained after the causal identification reflects this influence and, thus, correctly predicts the movement of $x_1$.

Only when lowering the threshold parameter can SINDy recover the true causal structure of the system. This stresses the general shortcoming of sparse identification methods when identifying the causal structure of a control system. Depending on the parameter settings, they may or may not recover the true causal structure. However, their general purpose is different. They seek a trade-off between model complexity and accuracy. Thus, neglecting a causal link with a comparably weak influence might be the desired outcome. In contrast, we seek the true causal structure, independent of how strong the influence is.

---

[4]We use the implementation provided in de Silva et al. (2020).

Second, we compare our algorithm to the PCMCI[5] algorithm proposed in Runge et al. (2019b)[6]. This algorithm focuses on detecting causal influences from time-series data. That is, it does not yield an identified system matrix but discovers which variables have a causal influence on which others. When again exciting the linear system with a chirp signal and running the PCMCI algorithm, we receive an output that we can interpret as

$$
\begin{aligned}
x_1(k+1) &= a_1 x_1(k) + a_2 x_2(k) + a_3 x_3(k) + b_1 u_1(k) \\
x_2(k+1) &= a_4 x_1(k-1) + a_5 x_2(k) + a_5 x_3(k) + a_6 x_3(k-2) + b_2 u_2(k) + b_3 u_3(k-1) \\
x_3(k+1) &= a_7 x_1(k-4) + a_8 x_3(k) + b_4 u_1(k) + b_5 u_3(k),
\end{aligned}
\tag{15}
$$

where all weights $a_i$ and $b_i$ are non-zero. Besides, the algorithm discovers an influence of $x_1$ on $u_3$. However, since such links were ruled out by design for the other algorithms, we neglect this here. Nevertheless, also the equations as stated above are not in line with the actual matrices in equation 11. For instance, following equation 15, the variable $x_1$ has a causal influence on $x_2$ and $x_3$ through the non-zero factors $a_4$ and $a_7$. However, given the upper-triangular structure of the $A$ matrix, $x_3$ is not influenced by any other state variable apart from itself. Similarly, also $a_4$ should be zero. Thus, the algorithm does not reveal the true structure of the system.

## 7 Conclusion

We presented a method that identifies the causal structure of dynamical control systems by conducting experiments and analyzing the generated data with MMD-based techniques. It differs from prior approaches to causal inference in that it uses a controllability notion that is suitable to design experiments for control systems. We evaluated the method on a real-world robotic system and a simulated quadruple tank system. Our algorithm successfully identified the underlying causal structure of both systems, which in turn allowed us to learn a model that accurately generalizes to previously unseen states while reducing computational complexity.

### Acknowledgments

The authors would like to thank Manuel Wüthrich and Alonso Marco Valle for insightful discussions, Steve Heim for valuable feedback, and Vincent Berenz for support with the robot experiments. This work has received funding from the German Research Foundation within the SPP 1914 (grant TR 1433/1-1), the Federal Ministry of Education and Research (BMBF) and the Ministry of Culture and Science of the German State of North Rhine-Westphalia (MKW) under the Excellence Strategy of the Federal Government and the Länder, the Cyber Valley Initiative, the Max Planck Society, the Knut and Alice Wallenberg Foundation, and the Swedish Research Council.

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

## A    Proof of Theorem 2

An LTI system with Gaussian noise follows a normal distribution, whose mean and variance are given by

$$\mathbb{E}[x(t)] = A^t x(0) + \sum_{i=0}^{t-1} A^i B u(t-1-i) \tag{16a}$$

$$\mathrm{Var}[x(t)] = \sum_{i=0}^{t-1} A^i \mathrm{Var}[v(t-1-i)], \tag{16b}$$

where we assume $\mathrm{Var}[v(t)] = \Sigma_{\mathrm{v}}$ for all $t$. For such systems, we will first show that if equation 16a obeys the controllability conditions stated by Kalman, the system is also controllable according to definitions 2 and 4.

**Lemma 3.** *The system in equation 6 is completely $\epsilon$-controllable in distribution if the deterministic part obeys the controllability condition stated in Kalman (1960a).*

*Proof.* The expected value in equation 16a represents the deterministic part of the system. Thus, according to Kalman (1960a), we can design an input trajectory that steers equation 16a to any point in the state space. Since we do not assume constraints on the input or the state variables, the desired state can be reached with a trajectory within $q < \infty$ time steps (cf. deadbeat control (Kalman, 1960b; Emami-Naeini & Franklin, 1982)). That is, starting from any $x(0) \in \mathcal{X}$ we can steer the system state toward $x(q) \overset{!}{=} x^{\mathrm{I}}$ in $q$ time steps and obtain the distribution

$$\mathbb{E}[x(q)] = x^{\mathrm{I}}$$

$$\mathrm{Var}[x(q)] = \sum_{i=0}^{q-1} A^i \Sigma_{\mathrm{v}}.$$

The probability of $\left\| x(q) - x^{\mathrm{I}} \right\|_2^2$ being larger than $\epsilon$ is given by the cumulative distribution function of the normal distribution. $\qquad\square$

**Proof of theorem 2.** We can now prove theorem 2. Since the variance of equation 6 solely depends on the number of time steps, which is equal for all experiments, distributions can only be different because of their means. We start with experiments that are designed according to equation 2a. In this case, for distributions to be equal, and, thus, for variables to be non-causal, we need

$$
e_i\left( A^t x^{\mathrm{I}}(0) + \sum_{i=0}^{t-1} A^i (B u(t-1-i)) \right) = e_i\left( A^t x^{\mathrm{II}}(0) + \sum_{i=0}^{t-1} A^i (B u(t-1-i)) \right),
$$

where $e_i \in \mathbb{R}^n$ is the unit vector (i.e., a vector with zeros and a single 1 at the $i$th entry). Since input trajectories are equal, this boils down to

$$
e_i A^t x^{\mathrm{I}}(0) = e_i A^t x^{\mathrm{II}}(0).
$$

Essentially this means that the component $ij$ of $A^t$ needs to be 0. This is clearly the case, if there is no influence of $x_i$ on $x_j$, i.e., in case variables are non-causal, we have MMD $= 0$. The event of component $ij$ of $A^t$ being 0 by chance, even though $x_j$ has a causal influence on $x_i$, has probability 0. Thus, we have that variables are non-causal if MMD $= 0$.

For experiments that are designed according to equation 2b, initial states are equal and, in case variables are non-causal, we have

$$
e_i \sum_{i=0}^{t-1} A^i (B u^{\mathrm{I}}(t-1-i)) = e_i \sum_{i=0}^{t-1} A^i (B u^{\mathrm{II}}(t-1-i)).
$$

Similar as before, we have equal distributions and, thus, MMD $= 0$ if entries in the $A^i B$ matrices relating $x_i$ and $u_j$ are 0, i.e., if there is no causal influence. The other direction holds since the event of the relevant entries being 0 by chance has probability 0.

## B    Further Example for Assumption 1

We provide a further example to empirically validate assumption 1. In practice, regions of local non-causality are often due to hysteresis effects. For instance, a resting body first needs to overcome static friction. That is, when the velocity is zero, the input signal needs to overcome a threshold to actually influence the velocity. In particular, we consider the system

$$
x(k+1) = \begin{cases} \begin{pmatrix} 1 & 0.01 \\ 0 & 1 \end{pmatrix} x(k) + \begin{pmatrix} 0 \\ 0.1 \end{pmatrix} u(k) + v(k) & \text{if } x_1(k) \neq 0 \vee |u(k)| > 0.1 \\ x(k) & \text{else,} \end{cases} \tag{17}
$$

where the state is 2-dimensional, the input is scalar, and $v(k)$ is a normally distributed random variable with standard deviation $1 \times 10^{-4}$. We excite the system with a ramp function starting at zero and going until two and learn a GP model. For the GP model, we use a Gaussian kernel for which we optimize the hyperparameters. We then assume the initial condition of both state variables to be zero and compute the MMD of simulated trajectories for 100 step inputs, with $u^{\mathrm{I}}$ ranging from 0.05 to 5 and $u^{\mathrm{II}} = -u^{\mathrm{I}}$. The results confirm our findings from example 1 since also here assumption 1 is satisfied (see figure 8).

## C    Further Results of the Robot Experiments

We first provide implementation details for the experiments presented in section 6. The initial model estimate is obtained by exciting the system with a chirp signal for $30\,\mathrm{s}$ and using the generated data to learn a linear state-space model (cf. equation 6) with least squares. The obtained matrices are

$$
A_{\mathrm{init}} \approx \begin{pmatrix} 0.868 & -0.132 & 0.754 & -0.491 \\ -0.132 & 0.868 & 0.754 & -0.491 \\ -0.132 & -0.132 & 1.754 & -0.491 \\ -0.134 & -0.134 & 0.76 & 0.508 \end{pmatrix} \quad B_{\mathrm{init}} \approx \begin{pmatrix} 0.075 & -0.056 & -0.031 & 0.022 \\ 0.074 & -0.055 & -0.031 & 0.022 \\ 0.074 & -0.056 & -0.03 & 0.022 \\ 0.075 & -0.056 & -0.032 & 0.022 \end{pmatrix}.
$$

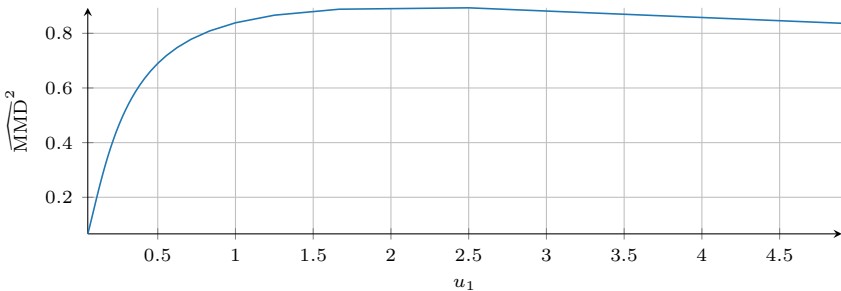

Figure 8: Simulated MMD for the hysteresis system in equation 17. *Similar as for example 1, also here we see that assumption 1 is satisfied.*

Initial conditions and input trajectories for the causality testing experiments are obtained through sophisticated guesses, as discussed in section 5.2. The found initial conditions and input trajectories yield expected MMDs orders of magnitude above the system's noise level for all joints. Thus, we need only eight experiments to identify the causal structure. We design input trajectories of 100 time steps for each experiment, repeat the experiment ten times, and use collected data from all experiments for hypothesis testing. While the squared MMD is always positive, the empirical approximation in equation 7 can become negative since it is an unbiased estimate. For the test statistic in equation 9, we estimate the variance using 100 Monte Carlo simulations and obtain the expected value through a noiseless simulation. We use $\nu = 1$, but as we will see in the results, the empirical MMD is, in all cases, orders of magnitude below or above the threshold. Thus, more conservative choices of $\nu$ would yield the same outcome.

In tables 1 and 2, we present the results of all causality testing experiments conducted on the robotic platform shown in figure 3. As for the results discussed in section 6, we always have a clear decision on whether to accept or reject the null hypothesis: The MMD found in experiments is always orders of magnitude larger or smaller than the test statistic. Also here, we find that all joints can be moved independently of each other and are affected by exactly one input. When exploiting the revealed causal structure for identifying the system matrices, we obtain

$$A_{\text{caus}} = \begin{pmatrix} 1 & 0 & 0 & 0 \\ 0 & 1 & 0 & 0 \\ 0 & 0 & 1 & 0 \\ 0 & 0 & 0 & 1 \end{pmatrix} \quad B_{\text{caus}} \approx \begin{pmatrix} 0.013 & 0 & 0 & 0 \\ 0 & 0.007 & 0 & 0 \\ 0 & 0 & 0.01 & 0 \\ 0 & 0 & 0 & 0.01 \end{pmatrix}.$$

## D   Further Results of the Quadruple Tank Experiments

We discretize the quadruple tank system with a time-step of $100\,\text{ms}$. We choose $A_i = 50\,\text{cm}^2$ for all tanks, $a_{1,2} = 0.242\,\text{cm}^2$, $a_{3,4} = 0.242\,\text{cm}^2$, and the valve parameters $\zeta_{1,2} = 0.5$. For the initial model learning, we excite the system for 5000 time steps. During excitation, the input is sampled from a univariate distribution with an interval $[0, 60]$. We use the generated data to learn a GP with a Gaussian kernel for each state variable. Having identified a model, we follow algorithm 1 to identify the causal structure. As for the robot experiments, we repeat each experiment 10 times and use 50 simulations to estimate the standard deviation. We choose $\nu = 10$ in equation 9 to avoid false positives. The results are collected in tables 3 and 4.

Table 1: Results of the causal structure identification for a robot arm. *Causal influences of joints on each other.*

| Joint $\rightarrow$ Joint | Experimental MMD | Test statistic |
|---|---|---|
| $x_1 \rightarrow x_1$ | 0 | $1.65 \times 10^{-4}$ |
| $x_1 \rightarrow x_2$ | 0 | $1.79 \times 10^{-4}$ |
| $x_1 \rightarrow x_3$ | 0 | $2.39 \times 10^{-4}$ |
| $x_1 \rightarrow x_4$ | $2.43 \times 10^{-13}$ | $1.61 \times 10^{-4}$ |
| $x_2 \rightarrow x_1$ | 0 | $5.6 \times 10^{-7}$ |
| $x_2 \rightarrow x_2$ | $-2.8 \times 10^{-18}$ | $4.58 \times 10^{-7}$ |
| $x_2 \rightarrow x_3$ | 0 | $3.56 \times 10^{-7}$ |
| $x_2 \rightarrow x_4$ | $1.82 \times 10^{-13}$ | $6.91 \times 10^{-7}$ |
| $x_3 \rightarrow x_1$ | 0 | $5.81 \times 10^{-7}$ |
| $x_3 \rightarrow x_2$ | 0 | $4.54 \times 10^{-7}$ |
| $x_3 \rightarrow x_3$ | $-1.38 \times 10^{-18}$ | $6.16 \times 10^{-7}$ |
| $x_3 \rightarrow x_4$ | $1.29 \times 10^{-16}$ | $5.2 \times 10^{-7}$ |
| $x_4 \rightarrow x_1$ | 0 | $4.99 \times 10^{-7}$ |
| $x_4 \rightarrow x_2$ | 0 | $4.66 \times 10^{-7}$ |
| $x_4 \rightarrow x_3$ | $9.63 \times 10^{-15}$ | $5.8 \times 10^{-7}$ |
| $x_4 \rightarrow x_4$ | $-5.44 \times 10^{-15}$ | $5.8 \times 10^{-7}$ |

Table 2: Results of the causal structure identification for a robot arm. *Causal influences of inputs on joints.*

| Input $\rightarrow$ Joint | Experimental MMD | Test statistic |
|---|---|---|
| $u_1 \rightarrow x_1$ | 0.04 | $1.18 \times 10^{-5}$ |
| $u_1 \rightarrow x_2$ | 0 | $5.38 \times 10^{-7}$ |
| $u_1 \rightarrow x_3$ | 0 | $6.51 \times 10^{-7}$ |
| $u_1 \rightarrow x_4$ | 0 | $4.47 \times 10^{-7}$ |
| $u_2 \rightarrow x_1$ | 0 | $5.09 \times 10^{-5}$ |
| $u_2 \rightarrow x_2$ | 0.04 | $1.15 \times 10^{-5}$ |
| $u_2 \rightarrow x_3$ | $3.51 \times 10^{-14}$ | $5.95 \times 10^{-7}$ |
| $u_2 \rightarrow x_4$ | $3.51 \times 10^{-14}$ | $4.67 \times 10^{-7}$ |
| $u_3 \rightarrow x_1$ | $2.31 \times 10^{-13}$ | $5.26 \times 10^{-5}$ |
| $u_3 \rightarrow x_2$ | $1.4 \times 10^{-13}$ | $4.28 \times 10^{-5}$ |
| $u_3 \rightarrow x_3$ | 0.04 | $1.18 \times 10^{-5}$ |
| $u_3 \rightarrow x_4$ | $2.25 \times 10^{-12}$ | $4.36 \times 10^{-7}$ |
| $u_4 \rightarrow x_1$ | 0 | $4.47 \times 10^{-5}$ |
| $u_4 \rightarrow x_2$ | 0 | $5.11 \times 10^{-5}$ |
| $u_4 \rightarrow x_3$ | 0 | $5.69 \times 10^{-7}$ |
| $u_4 \rightarrow x_4$ | 0.04 | $6.58 \times 10^{-4}$ |

Table 3: Results of the causal structure identification for a quadruple tank system. *Causal influences of the tanks on each other.*

| Tank → Tank | Experimental MMD | Test statistic |
|---|---|---|
| $x_1 \to x_1$ | $2.78 \times 10^{-1}$ | $9.81 \times 10^{-4}$ |
| $x_1 \to x_2$ | $7.39 \times 10^{-6}$ | $9.55 \times 10^{-5}$ |
| $x_1 \to x_3$ | $1.47 \times 10^{-5}$ | $9.78 \times 10^{-5}$ |
| $x_1 \to x_4$ | $9.45 \times 10^{-5}$ | $1.01 \times 10^{-3}$ |
| $x_2 \to x_1$ | $3.57 \times 10^{-6}$ | $2.35 \times 10^{-5}$ |
| $x_2 \to x_2$ | $2.74 \times 10^{-1}$ | $4.58 \times 10^{-4}$ |
| $x_2 \to x_3$ | $2.32 \times 10^{-6}$ | $4.31 \times 10^{-6}$ |
| $x_2 \to x_4$ | $3.54 \times 10^{-5}$ | $1.41 \times 10^{-4}$ |
| $x_3 \to x_1$ | $1.62 \times 10^{-1}$ | $1.29 \times 10^{-4}$ |
| $x_3 \to x_2$ | $5.36 \times 10^{-6}$ | $3.28 \times 10^{-4}$ |
| $x_3 \to x_3$ | $3.44 \times 10^{-1}$ | $2.75 \times 10^{-4}$ |
| $x_3 \to x_4$ | $9.59 \times 10^{-6}$ | $3.46 \times 10^{-5}$ |
| $x_4 \to x_1$ | $4.28 \times 10^{-6}$ | $4.04 \times 10^{-5}$ |
| $x_4 \to x_2$ | $1.45 \times 10^{-1}$ | $8.74 \times 10^{-5}$ |
| $x_4 \to x_3$ | $2.35 \times 10^{-6}$ | $6.56 \times 10^{-6}$ |
| $x_4 \to x_4$ | $3.57 \times 10^{-1}$ | $1.42 \times 10^{-4}$ |

Table 4: Results of the causal structure identification for a quadruple tank system. *Causal influences of inputs on tanks.*

| Input → Tank | Experimental MMD | Test statistic |
|---|---|---|
| $u_1 \to x_1$ | $1.8 \times 10^{-1}$ | $1.01 \times 10^{-4}$ |
| $u_1 \to x_2$ | $8.04 \times 10^{-3}$ | $8.74 \times 10^{-5}$ |
| $u_1 \to x_3$ | $2.46 \times 10^{-8}$ | $3.8 \times 10^{-5}$ |
| $u_1 \to x_4$ | $1.56 \times 10^{-1}$ | $8.61 \times 10^{-5}$ |
| $u_2 \to x_1$ | $7.55 \times 10^{-3}$ | $9.86 \times 10^{-5}$ |
| $u_2 \to x_2$ | $1.78 \times 10^{-1}$ | $1.12 \times 10^{-4}$ |
| $u_2 \to x_3$ | $1.58 \times 10^{-1}$ | $3.77 \times 10^{-5}$ |
| $u_2 \to x_4$ | $5.44 \times 10^{-6}$ | $4.46 \times 10^{-5}$ |

