# OpenReview forum: "Identifying Causal Structure in Dynamical Systems"
_TMLR — Accepted by TMLR_

### Review · Reviewer_QpEU · 2022-04-22

**Summary Of Contributions:**

In this paper, the authors study the problem of identifying causality in dynamical systems. To this end, they introduce an algorithm based on measuring maximum mean discrepancy. Implicitly, the method considers controllability, making it a viable method to identify causality in control system. The method is evaluated in simulations consisting of a robotic arm and a quadruple tank process.

**Broader Impact Concerns:**

None.

**Requested Changes:**

Addressing the previously mentioned weaknesses should be sufficient.

**Strengths And Weaknesses:**

Strengths. This is a work of possible interest to the community. Writing and presentation are clear and, for the most part, concepts and statements are clearly introduced and treated in a rigorous manner. The idea introduced in the paper is simply yet well founded.

Weaknesses.

(i) I think that knowledge of do-calculus might not be that common among readers. Doing a soft introduction to it might be beneficial to the paper. The do-operator appears in Definition 1 without much introduction (even though it is explained afterwards). Maybe a "Preliminaries" section would help the reader more unfamiliar with do-calculus.

(ii) The related work section and the overall framing of this work with respect of the state of the art is somewhat odd. Most of the works cited, while important works, are quite old. While reading the manuscript I had a hard time placing this work with respect to more recent works addressing the problem of identifying causality. Overall, it feels it feels like a work disconnected from recent work. I'd suggest a better reframing of the Related Work section to put better emphasis on how this work relates to recent state of the art.

(iii) Why is the maximum mean discrepancy used as the measure of similarity (instead of something like the KL divergence). This is not clearly explained/motivated. A somewhat vague justification is given for the case of reproducing kernel Hilbert spaces, but I'd suggest clarifying this further.

(iv) Overall, the information presented in the numerical results is simplistic. For example, a motivation for learning causality is to reduce model complexity, but this is not really evaluated in the numerical results. In the same sense, for the quadruple tank process, only the causality tests are shown (and extended in the Appendix). However, it would be interesting to see how a reduced model learned from the causality learning process behaves. I'd suggest extending the numerical results with more insightful figures and discussion.

---

> ### Author Response · Authors · 2022-05-08
> **Replies to review**
>
> We thank the reviewer for the positive evaluation of our paper.
>
> We thank the reviewer for pointing out the missing theoretical background on the do-calculus. As discussed in the answer to Reviewer 7Puu, we have now in the revised version included some background on the do notation.
>
> To address the comment on our related work section, we added some more recent relevant works (e.g., Runge et al., Nature Communications, 2019, Salvi et al., NeurIPS, 2021). We prefer to keep the old references as they illustrate how different scientific communities have considered similar problem settings as the one we consider in this paper. Thus, in the related work section, we aim at providing a brief overview of these different approaches. We find that the problem of identifying causal structure in dynamical systems considering a proper notion of controllability has not been addressed before.
>
> The maximum mean discrepancy (MMD) is indeed one of many possibilities for measuring the difference between trajectories. Another popular choice is the KL divergence. An overview of such probability metrics is provided by Sriperumbudur et al., "On the empirical estimation of integral probability metrics," Electronic Journal of Statistics, 2012. Overall, the MMD has the advantages that we can estimate it without explicitly estimating the probability distributions, that it is easy to calculate, and it provides theoretical guarantees for correctly deciding whether or not two probability distributions are equal without requiring many assumptions. We have added Remark2 in the revised version that discusses different metrics that could be chosen and motivates our choice of the MMD.
>
> We thank the reviewer for the comment on our experimental evaluation. As mentioned in the answer to Reviewer VmF2, we have now expanded the experimental section by including a discussion on the choice of hyperparameters and a comparison with a causal discovery algorithm. We would also like to point out that for the robot arm and the synthetic example, we demonstrate the enhanced generalization capabilities of the model learned after identifying the causal structure. Regarding the Gaussian process model of the quadruple tank process, we here see the advantage mainly in the reduced computational complexity when querying or relearning the system model. The complexity of Gaussian process regression scales cubically with the number of data points. Thus, if we can decrease the model dimension from ten to three, we reduce the number of data points to be considered by 70%, which greatly reduces the computational complexity. We have expanded the discussion in the revised version to more clearly point out this important advantage.

---

### Review · Reviewer_VmF2 · 2022-04-27

**Summary Of Contributions:**

This paper presents an experimental design method for causal structure identification in general, possibly non-linear dynamical systems which are not necessarily fully controllable. The method is based on comparing the trajectory distributions under different settings for initial conditions and control signals. Theoretical results are derived relating the maximum mean discrepancy (MMD) of the trajectory distributions and causal relations within controllability limitations. A set of experiments demonstrating the method on a physical robot and two synthetic problems is also presented.

**Broader Impact Concerns:**

No major broader impact concern, other than perhaps adding a brief discussion on the possible consequences of inferring causality in experiments outside of robotic settings, which may involve human agents, for example, case that's a suitable and relevant application.

**Requested Changes:**

Considering the issues presented above, I suggest:
- 1. Adding a brief theoretical background on do-calculus in the context of causal identification
- 2. Revising the notation in the main theoretical results
- 3. Revising theoretical statements, making sure that notation is properly explained and formally defined
- 4. Adding a discussion on hyper-parameters tuning
- 5. Ablation experiments showing how the method behaves as a function of hyper-parameter settings
- 6. Adding further experimental baselines to experiments in Sec. 6.

**Strengths And Weaknesses:**

The paper addresses an important problem on causality in the context of control systems. As the paper discusses, most causal identification algorithms are either entirely based on observational data or interventions where one has full control over the variables of interest. Therefore, I believe that an approach that takes controllability restrictions into account is interesting for the machine learning community working on causality. However, the paper also comes with a number of issues which make me believe it's not ready yet for publication.

#### Major issues:
- 1) **Insufficient theoretical background.** The proposed method is based on Judea Pearl's do-calculus, but no (minimal) background on do-calculus is provided. Although one could argue that is possible to understand most of the paper without this background, the problem setting in Sec. 3.1 gives the impression that it is actually a somewhat necessary background.

- 2) **Notation issues.** The notation in the paper can be quite confusing sometimes, as some of the symbols have their definition coming late or are left implicit/unexplained. For instance, the subscripts and superscripts in Definition 1 have no prior definition. Later on, it can be *assumed* that the subscript $i$ on a state variable $x_i$ denotes the $i$th coordinate of that variable, or perhaps an individual vector-valued variable within a group of vector-valued variables composing the state of the system (the paper does not make it clear). Nowhere in the paper I have seen a formal specification for the control and state spaces. They're only left to be assumed as some possibly arbitrary vector space of some dimensionality. It is also not clear what the superscripts "I" and "II" represent in Definition 1 in a general context. Furthermore, $\mathcal{X}_{nc}$ and the corresponding $\mathcal{U}$ are only loosely explained, but not formally defined.

- 3) **Confusing theoretical statements.** Assumption 1 sounds more like a theorem than an assumption due to the last sentence in its statement. It might need to be split into assumption and theorem/lemma/remark statements, depending on whether it requires a proof or the proof happens to be trivial.

- 4) **Hyper-parameters.** The proposed method has a few hyper-parameters, such as the MMD thresholds $\delta_1$ and $\delta_2$, but the paper apparently does not discuss how these hyper-parameters can be tuned or set accordingly in practice for different applications.

- 5) **Lack of experimental baselines.** The experiments section does not provide comparisons against other suitable algorithmic baselines, other than  SINDy in the last synthetic experiment. Although existing work is mostly based on observational data or full interventions, it would be interesting to see how the proposed algorithm fares against other methods in settings where they can be applied. As far as I understand, at least the methods based on purely observational data can be applied to the same problems explored in the paper. An interesting question would then be whether they can identify the same causal relationships.

Minor issues:
- I believe there's a typo in the definition of $m_2$ in Lemma 2.
- Kinematic model in experiment 1. Torque is proportional to angular acceleration, not speed, as in the equation $\dot{\phi}_i = \tau_i$, if $\phi_i$ indeed represents angular position/angle.

---

> ### Author Response · Authors · 2022-05-08
> **Replies to review**
>
> We thank the reviewer for the thorough review of our paper.
>
> We thank the reviewer for pointing out the missing theoretical background on the do-calculus. As discussed in the answer to Reviewer 7Puu, we have now in the revised version included some background on the do notation.
>
> We also thank the reviewer for pointing out the notation issues. We have addressed all of them in a revised version. Note that in Section 3, we define the state and control spaces to be n- and m-dimensional vector spaces, respectively. We have reformulated the last sentence of Assumption 1 to clarify that it is actually an assumption.
>
> We thank the reviewer for suggesting more discussion on the chosen hyperparameters. The hyperparameters $\delta_1$ and $\delta_2$ should be chosen in accordance with the noise level of the system. That is, to be sure that we are outside of regions of local non-causality, the MMD should be significantly above the noise level of the system. Since we estimate a model of the system in the beginning, we also have an estimate of that noise level. For our experiments, we use sophisticated guesses to choose the initial conditions and input trajectories. These already lead to empirical MMDs that are above the noise level of the system. Thus, lowering the $\delta_1$ and $\delta_2$ thresholds would not result in a different behavior of the algorithm. If the thresholds would be raised, we would need to further optimize the initial conditions and trajectories before starting the experiments. To illustrate this, we - in the revised version - report empirical MMD values and discuss how different choices of $\delta_1$ and $\delta_2$ would affect the algorithm for the robot experiments.
>
> As a response to the comment on missing experimental baselines, we tested the causal discovery algorithm proposed in Runge et al., "Detecting and quantifying causal associations in large nonlinear time series datasets," Science Advances, 2019, with the synthetical example. Given 100'000 data points from the system that was excited with a chirp signal, the algorithm was not able to correctly identify the causal graph, similar as the SINDy algorithm. We have included this comparison in the revised version of the paper. Generally, also causal discovery algorithms that target time series data or dynamical systems should be able to correctly identify the graph. However, they depend on appropriate data while our algorithm utilizes the input of a dynamical system to actively create the data needed for causal inference.

---

### Review · Reviewer_7Puu · 2022-04-29

**Summary Of Contributions:**

The paper presents an algorithm for identifying causal structure of the dynamical systems, predicting which state/control dimensions are causally related and which are not. Determining causality is increasingly important in controlling dynamical systems as providing powerful inductive bias that would be hard to determine from data automatically via deep ML. The algorithm presented by the authors is based on the non-causality definition related to the one from Pearl 1995. The algorithm comes in two different variants: (a) the one where initial conditions vary whereas controls are thee same and (b) the one where initial conditions vary, but the controls differ. The method proposed by the authors uses maximum mean discrepancy (MMD) based metrics (measuring discrepancy between trajectories) to identify causal relationships and discuss how to apply them in the control setting, where states can be achieved only by applying a sequence of controls. In order to scale up the approach, the authors leverage kernel-based unbiased estimator of the squared MMD wit the use of the characteristic kernels. Presented theoretical results are accompanied with empirical evaluation on: (a) the kinematic control of the robot, (b) the quadruple tank process (to test the setting with highly nonlinear dynamics) and some synthetic tasks.

**Broader Impact Concerns:**

I do not see any concerns regarding ethical aspects of this work.

**Requested Changes:**

Strengthening the experimental section would be very welcome. The authors can also work on improving the presentation of the paper. For instance: (a) the Authors refer to "do-" notation in Definition 1, without properly introducing it earlier, (b) while providing an introduction to MMDs the authors refer to characteristic kernel as useful while working with the empirical MMD, but do not provide their definition (just Gaussian kernel as an example). Most importantly, the Authors should comment on the Assumption 1 which is critical for the correctness of the algorithm (the proof that the proposed method is almost immediate if we use it).

**Strengths And Weaknesses:**

The proposed algorithm (formulations: 5a, 5b) is an elegant application of the MMD methods and the authors do a good job in explaining how to make this approach scalable. What I found very important is the discussion how to achieve desired states for the presented algorithm by controlling the system which is crucial from the practical point of view (Section 4.2 on controllability). Presented empirical results show that the method is very effective in practice do determine causal relationships.

The weakness of the approach is that even though elegant, presented algorithm is pretty straightforward given used definition of non-causality combined with Assumption 1 (MMD formulation is almost tautologically implied by Definition 1 and Assumption 1). One of my concerns is that it is not clear at all which dynamical systems f satisfy Assumption 1 or how to test this efficiently. Given the questions regarding how constraining Assumption 1 is, it is not clear to me how widely applicable the methods presented by the Authors are.

The paper would also benefit from adding few more difficult and high-dimensional dynamical systems setting, in particular to test the scalability of the algorithm and its data efficiency.

---

> ### Author Response · Authors · 2022-05-08
> **Replies to review**
>
> We thank the reviewer for the positive evaluation of our paper.
>
> We thank the reviewer for pointing out the missing introduction to the do-notation. We have revised the paper and in the revised version we include a clearer definition of the do-notation before starting Section 3.1. Since the do-calculus is a large framework, we only introduce the notation to the extent that it is used in the paper.
>
> Characteristic kernels are widely used when working in reproducing kernel Hilbert spaces. For our setting, choosing a characteristic kernel ensures that the maximum mean discrepancy (MMD) is injective, i.e., only then we have MMD(P,Q) = 0 if and only if P = Q. Thus, the restriction to characteristic kernels is critical for our theoretical claims to hold. Many popular kernels, such as the Gaussian and the Laplacian kernels, are characteristic and, therefore, this restriction is not a limitation in practice. How to check whether a kernel is characteristic is quite technical and would not contribute to the main narrative of the paper. Hence, we prefer to leave this discussion out and instead refer to the literature.
>
> Assumption 1 is indeed critical for our algorithm. However, it is only critical for systems in which we have regions of local non-causality. As we discuss in Section 4.3, in linear systems, for instance, such regions do not exist and, hence, Assumption 1 is not required. That is, only for a subclass of systems is the assumption actually required (also nonlinear systems do not necessarily have regions of local non-causality). Assume we consider a system with regions of local non-causality. On an intuitive level, Assumption 1 says that if $x_1$ has no influence on $x_2$ in certain parts of the state/action space, our model may, due to spurious correlation, still assign some influence of $x_1$ on $x_2$ in those regions. However, we expect the model to assign a stronger influence in regions where the influence stems not only from spurious correlations, but also from an actual physical influence of $x_1$ on $x_2$. We think that for a suitable excitation and model class used to learn/identify the system model, this assumption is reasonable. Empirically, we verify it to be satisfied for the system considered in Example 1. Lastly, the assumption is actually stricter than needed. Since we introduce the maximization procedure in Equation (5), we make sure that we perform experiments in regions where we expect the MMD to be maximized. Thus, for the theoretical claim not to hold, we would need that our system model expects the MMD to be at its maximum in regions of local non-causality. This we consider a very unlikely case to happen in practice. To address the reviewer's concern, we have added more discussion on Assumption 1 and introduced Remark 3.

---

> > ### Comment · Reviewer_7Puu · 2022-06-06
> > **Thank you for your response**
> >
> > I thank the Authors very much for their comments.  Unfortunately I am still not fully convinced by the Authors' responses. The main problem is Assumption 1. The Authors admit that it might be not satisfied in some nonlinear systems with regions of local non-causality, but they do not provide even a rough simpler characterization of such systems or how to efficiently test whether a given system is of that type. The Authors claim that in practice the assumption holds. Unfortunately this is true only for the limited set of experiments currently presented in the paper. Suggestions were made to extend the experimental scope (that would at least empirically test the conjecture that Assumption 1 is in practice very weak). To the best of my knowledge, those additional experiments were not conducted. Also, relying theoretical analysis on Assumption 1 makes the theoretical results a little less interesting since they are a straightforward consequence of this assumption.
> >
> > Thank you for your comment on the characteristic kernels. This is a standard tool in MMD. What I meant here is to define the term if it is being introduced. This is of course a minor comment.
> >
> > To summarize, this is an interesting paper with the *mathematically elegant approach to the problem*. Thee concern though is that it is not clear how practical it is given the above discussion.

---

> > > ### Author Response · Authors · 2022-06-08
> > > **Reply**
> > >
> > > We thank the reviewer for this comment. Since Assumption 1 is indeed an important ingredient of our algorithm, we have now added another example which is included in the new revised version. In this example, we consider a system with a hysteresis. An intuitive example for such a system would be an autonomous car which, when standing still, first has to overcome static friction to start moving. I.e., when starting with a velocity of zero, low input signals have no effect, which may lead to the wrong conclusion that the input generally has no causal influence on the velocity. We learn a GP model for this system and show that the simulated MMD is indeed minimal for low input signals. That is, Assumption 1 is satisfied and our experiment design would yield experiments that reveal the causal influence of the input on the velocity.
> > >
> > > We would like to stress that we did not say in our response that Assumption 1 may not be satisfied for certain nonlinear systems. What we said was that only certain nonlinear systems actually have regions of local non-causality - that is, for many systems (including all linear systems), Assumption 1 is not needed at all. For the systems that have regions of local non-causality (e.g., systems with a hysteresis), our claim is that in practice, Assumption 1 is typically satisfied, for which we now provide further evidence with this new example.

---

> > > > ### Comment · Reviewer_7Puu · 2022-06-22
> > > > **response**
> > > >
> > > > Dear Authors,
> > > >
> > > > I'd like to sincerely thank you for your comments. System with the hysteresis is an interesting example, yet my concern is that the main question: more general characterization of the nonlinear systems where the algorithm can be applied is not known. Thus it is very hard to assess how practical the algorithm is. Extending the experimental section and adding more experimental results with nonlinear systems (if it is hard to derive more general theory) would certainly help but to the best of my knowledge, the empirical evaluation has not been extended. Thus I do not change the score.

---

> > > > > ### Author Response · Authors · 2022-06-27
> > > > > **reply**
> > > > >
> > > > > We thank the reviewer for the response. We would like to stress that we did strengthen the numerical evaluation in the previous revision. The new example (the system with hysteresis) was added in the appendix and not in the evaluation section, but is still a part of the numerical evaluation of our method. Overall, with all due respect, we think that we have provided clear evidence that our algorithm is applicable to a wide class of systems. We have successfully evaluated it in important system identification benchmarks and even in a hardware experiment. In particular, there is no evidence that there exists a class of systems for which our algorithm would not be applicable. A set of relevant benchmark systems is provided here: https://www.nonlinearbenchmark.org/benchmarks. This list of systems contains a robot arm, a multi tank system, and a system with hysteresis - all of which is part of our evaluation section. Further, the only examples of local non-causalities that we could find in the provided systems were hysteresis effects - for which we have demonstrated that our algorithm can design experiments that still allow one to identify the true causal structure. Thus, our algorithm is applicable to all systems in this benchmark.
> > > > >
> > > > > The reviewer has mentioned that it was suggested to further strengthen the evaluation section. In the course of the review process, we have significantly strengthened the evaluation by adding a further example system (the system with hysteresis), by extending the discussion on saving computational power through our algorithm for the tank system, and by adding a comparison with a causal discovery method. Unfortunately, it is not clear to us which additional experiments would be desirable.

---

### Decision · Action_Editors · 2022-06-27

**Recommendation:** Accept with minor revision

**Comment:**

Learning the causal structure of an unknown dynamical system and exploiting it for data-efficient control is a very well motivated and important problem, with many applications of cross-disciplinary interest. This paper develops a class of MMD based methods to compare trajectory distributions under varying initial conditions/control sequences to infer causality, which implicitly accounts for controllability of the underlying system -- the reviewers agree this is an elegant approach overall, presented clearly and rigorously in the paper. Several concerns and issues were raised around comprehensiveness of the experimental results, but they were diligently addressed during the review period and the AE believes the expansions described in the author responses are sufficient. For final revision, all reviewers requested a more complete and self-contained introduction to "do-calculus", related work that better covers more recent works, and a stronger commentary around what classes of nonlinear systems may or may not be captured by Assumption 1.